# To Believe or Not to Believe Your LLM: Iterative Prompting for Estimating Epistemic Uncertainty

**Yasin Abbasi Yadkori**
Google DeepMind
yadkori@google.com

**Ilja Kuzborskij**
Google DeepMind
iljak@google.com

**András György**
Google DeepMind
agyorgy@google.com

**Csaba Szepesvári**
Google DeepMind and University of Alberta
szepi@google.com

## Abstract

We explore uncertainty quantification in large language models (LLMs), with the goal to identify when uncertainty in responses given a query is large. We simultaneously consider both epistemic and aleatoric uncertainties, where the former comes from the lack of knowledge about the ground truth (such as about facts or the language), and the latter comes from irreducible randomness (such as multiple possible answers). In particular, we derive an information-theoretic metric that allows to reliably detect when only epistemic uncertainty is large, in which case the output of the model is unreliable. This condition can be computed based solely on the output of the model obtained simply by some special iterative prompting based on the previous responses. Such quantification, for instance, allows to detect hallucinations (cases when epistemic uncertainty is high) in both single- and multi-answer responses. This is in contrast to many standard uncertainty quantification strategies (such as thresholding the log-likelihood of a response) where hallucinations in the multi-answer case cannot be detected. We conduct a series of experiments which demonstrate the advantage of our formulation. Further, our investigations shed some light on how the probabilities assigned to a given output by an LLM can be amplified by iterative prompting, which might be of independent interest.

## 1 Introduction

Language models too occasionally suffer from *hallucinations*, or responses with low truthfulness, that do not match our own common or textbook knowledge (Bubeck et al., 2023; Gemini Team, Google, 2023). At the same time, since LLMs work by modeling a probability distribution over texts, it is natural to view the problem of truthfulness through the lens of statistical uncertainty. In this paper we explore uncertainty quantification in LLMs. We distinguish between two sources of uncertainty: *epistemic* and *aleatoric* (Wen et al., 2022; Osband et al., 2023; Johnson et al., 2024). Epistemic uncertainty arises from the lack of knowledge about the ground truth (e.g., facts or grammar in the language), stemming from various reasons such as insufficient amount of training data or model capacity. Aleatoric uncertainty comes from irreducible randomness in the prediction problem, such as multiple valid answers to the same query. Hence, truthfulness can be directly analyzed via looking at the epistemic uncertainty of a model in the sense that when the epistemic uncertainty is low, the model predictions must be close to the ground truth.

38th Conference on Neural Information Processing Systems (NeurIPS 2024).

Rigorously identifying when (either) uncertainty is small[1] is notoriously hard, especially in deep neural networks (Blundell et al., 2015; Antorán et al., 2020). This is because we generally lack guarantees about learning the ground truth (consistency), or even a weaker guarantee about how large the variance of a learning algorithm is. At the same time, there exist many heuristic approaches for uncertainty quantification based on simply looking at the log-likelihood of responses (Kadavath et al., 2022), estimating entropy (Kuhn et al., 2023), ensembling (Lakshminarayanan et al., 2017b; Dwaracherla et al., 2023; Osband et al., 2023), or sometimes even more principled formulations, such as conformal prediction (Angelopoulos et al., 2023; Ravfogel et al., 2023; Yadkori et al., 2024) (which however come with strong assumptions).

To the best of our knowledge, a common limitation of these approaches is that they are only meaningful in problems where there exists a *single* correct response (e.g. label) as they aim for detecting if one response is dominant (or multiple responses with the same meaning), that is, if there is only little uncertainty in the prediction. On the other hand, when multiple responses are correct, that is, there is *aleatoric uncertainty* in the ground truth, simply estimating the amount of uncertainty in the LLM's output is insufficient, as the perfect (ground-truth) predictor may have large aleatoric uncertainty and no epistemic uncertainty, while a completely useless predictor may have large epistemic uncertainty only, but the total amount of uncertainty of the two predictors might be the same.

**Contributions.** In this paper we address the above problem directly, and design methods to *decouple epistemic and aleatoric uncertainty*, allowing us to effectively deal with multi-response queries. Rather than trying to quantify how small epistemic uncertainty can be, we aim to identify when only the *epistemic uncertainty is large*, in which case we can suspect that the response is hallucinated.[2]

As a starting point we make a simple observation: If multiple responses are obtained to the same query from the ground truth (the language), they should be independent from each other, that is, in probabilistic interpretation, the joint distribution of these multiple responses, for a fixed query, must be a product distribution.

This observation can be used to measure how *far* the language model can be from the ground truth. The sequential model implemented by a language model allows us to construct a joint distribution over multiple responses, which is done through *iterative prompting of an LLM based on its previous responses* and the application of the chain rule of probability: first the model is asked to provide a response given a query, then to provide another response given the query and the first response, then a third one given the query and the first two responses, an so on. This is in contrast to some of the earlier works that approached decoupling epistemic and aleatoric uncertainty for classification problems by training the model with label pairs (or tuples) (Wen et al., 2022; Johnson et al., 2024).

So, if the response to a prompt containing the query and previous responses is insensitive to the previous responses, we have the desired independence and the LLM-derived joint distribution can be arbitrarily close to the ground truth. On the other hand, if the responses within the context heavily influence new responses from the model then, intuitively speaking, the LLM has low confidence about the knowledge stored in its parameters, and so the LLM-derived joint distribution *cannot be close* to the ground truth. As more responses are added to the prompt, this dependence can be made more apparent, allowing to detect *epistemic uncertainty via our iterative prompting procedure*.

Interestingly, as we will see in Section 3, we can force an LLM to provide a desired (possibly incorrect) response by adding this response repeatedly to the prompt. This phenomenon is then further investigated from the viewpoint of a transformer LLM architecture in Section 4.

The iterative prompting procedure then leads to the following main contributions:

*(i)* Based on the above iterative prompting procedure, we derive an *information-theoretic metric of epistemic uncertainty* in LLMs (Section 5), which quantifies the gap between the LLM-derived distribution over responses and the ground truth. This gap is insensitive to aleatoric uncertainty, allowing to quantify epistemic uncertainty even in cases where there are multiple valid responses.

*(ii)* We derive a computable lower bound on this metric, which turns out to be a *mutual information* (MI) of an LLM-derived joint distribution over responses, and propose a finite-sample estimator for it.

---

[1]For instance, by saying that predictions live in a confidence set with high probability.

[2]In technical terms this corresponds to giving a lower bound, rather than an upper bound, on the quantity capturing the uncertainty.

We prove that this finite-sample MI estimator sometimes suffers only a negligible error even though LLMs and their derived joint distributions are defined over potentially infinite supports (all possible strings in a language).

*(iii)* We discuss an algorithm for hallucination detection based on thresholding a finite-sample MI estimator, where the threshold is computed automatically through a *calibration* procedure. We show experimentally on closed-book open-domain question-answering benchmarks (such as TriviaQA, AmbigQA, and a dataset synthesized from WordNet) that when the data is mostly composed of either single-label or multi-label queries, our MI-based hallucination detection method surpasses a naive baseline (which is based on the likelihood of the response), and achieves essentially similar performance to that of a more advanced baseline which is based on the entropy of the output as a proxy for uncertainty. However, on datasets which contain both single- and multi-label samples at the same time, our method also significantly outperforms the entropy-based baseline, by achieving a much higher recall rate on samples with high output entropy while maintaining similar error rates.

*(iv)* Focusing on a single self-attention head, we identify a simple mechanistic explanation for how the model output can be changed through iterative prompting using previous responses, as discussed earlier. Suppose that the prompt is composed from a query and a repeated element (e.g., a possibly wrong answer). If the query lies within the space spanned by the large principal components of a key-query matrix product, then the output will be generated according to the knowledge extracted from the training data (now stored in a value matrix). On the other hand, if the query has little overlap with the large principal components, then the repeated element is likely to be copied from the prompt.

## 2    Preliminaries

**Conditional distributions and prompting.**    Let $\mathcal{X}$ be the space of finite text sequences, that is $\mathcal{X} \subset \Sigma^*$ where $\Sigma$ is a finite alphabet (and $\Sigma^* = \bigcup_{n=1}^{\infty} \Sigma^n$). Moreover, consider a family of conditional distributions $\mathcal{P} = \{\mu : \mathcal{X} \to [0,1] \mid \sum_{x \in \mathcal{X}} \mu(x \mid x') = 1 \quad \forall x' \in \mathcal{X}\}$. In the following, we let $P \in \mathcal{P}$ be the ground-truth conditional probability distribution over text sequences (responses) given a prompt, and we let $Q \in \mathcal{P}$ be the learned language model. Given a fixed query $x \in \mathcal{X}$ and possible responses $Y_1, \ldots, Y_t$, we define a *family of prompts* $\mathcal{F} = \{F_t : \mathcal{X} \to \mathcal{X} \mid t \in \mathbb{N}\}$, such that $F_t(x, Y_1, \ldots, Y_t)$ is defined as:

> Consider the following question: Q: $x$
> One answer to question Q is $Y_1$. Another answer to question Q is
> $Y_2$.[. . .] Another answer to question Q is $Y_t$.
> Provide an answer to the following question:
> Q: $x$. A:

**Information-theoretic notions.**    Let $\mu, \mu'$ be distributions supported on set $\mathcal{Z} = \mathcal{Z}_1 \times \cdots \times \mathcal{Z}_n$ where $(\mathcal{Z}_i)_i$ is a collection of countable sets. The *entropy* of a distribution $\mu$ is defined as $H(\mu) = \sum_{z \in \mathcal{Z}} \mu(z) \ln(1/\mu(z))$.[3] If $\mu, \mu'$ are such that $\mu'(z) = 0$ only if $\mu(z) = 0$, we have a *Kullback-Leibler divergence* between them defined as $D_{\mathrm{KL}}(\mu, \mu') = \sum_{z \in \mathcal{Z}} \mu(z) \ln(\mu(z)/\mu'(z))$. For any $z \in \mathcal{Z}$, we denote $z^{\backslash i} = (z_1, \ldots, z_{i-1}, z_{i+1}, \ldots, z_n)$, and the marginal of the $i$th coordinate of $\mu$ is given by $\mu_i(z) = \sum_{z^{\backslash i} \in \mathcal{Z}^{n-1}} \mu(z)$. The product distribution of the marginals of $\mu$ is given by $\mu^{\otimes}(z) = \prod_{i=1}^n \mu_i(z)$, and the *mutual information* of $\mu$ is defined as $I(\mu) = D_{\mathrm{KL}}(\mu, \mu^{\otimes})$.

## 3    Probability amplification by iteratively prompting

In this section we demonstrate that, as mentioned in the introduction, repeating possible responses several times in a prompt can have pronounced effects on the output of a language model. Consider $x =$ *"What is the capital of the UK?"* and $Y_1 = \cdots = Y_t =$ *"Another answer to question Q is Paris."* Here we can repeat the sentence *"Another answer to question Q is Paris."* an arbitrary number of times. Although the number of repetitions changes the behavior of the LLM, the correct response maintains a significant probability: as Figure 1 shows, the conditional normalized probability[4] of

---

[3]Following the usual convention, we define $0 \ln 0 = 0$ and $a \ln(a/0) = \infty$ for any $a > 0$.

[4]To obtain conditional normalized probabilities, we consider the probabilities of the two responses, and normalize them so that they add to 1.

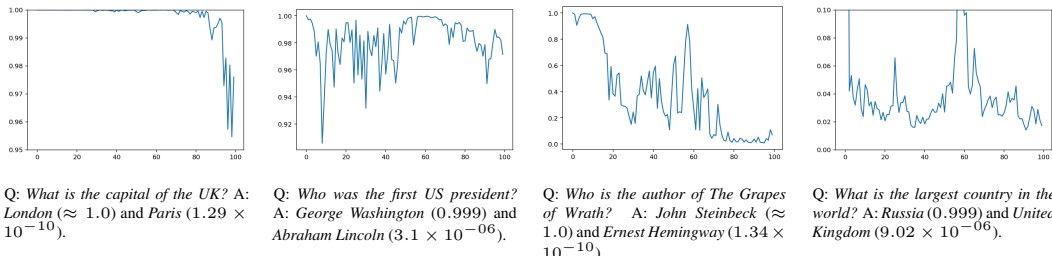

Q: *What is the capital of the UK?* A: *London* ($\approx$ 1.0) and *Paris* ($1.29 \times 10^{-10}$).

Q: *Who was the first US president?* A: *George Washington* (0.999) and *Abraham Lincoln* ($3.1 \times 10^{-06}$).

Q: *Who is the author of The Grapes of Wrath?* A: *John Steinbeck* ($\approx$ 1.0) and *Ernest Hemingway* ($1.34 \times 10^{-10}$).

Q: *What is the largest country in the world?* A: *Russia* (0.999) and *United Kingdom* ($9.02 \times 10^{-06}$).

Figure 1: Single-label queries with low epistemic uncertainty: Conditional normalized probability of the correct completion given repetitions of an incorrect response. Each figure shows the query and the considered two responses with their initial probabilities, as a response for the query, in parentheses (the first response is the correct one).

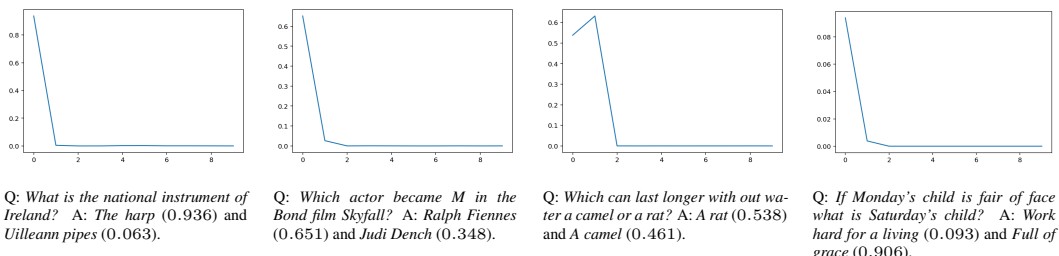

Q: *What is the national instrument of Ireland?* A: *The harp* (0.936) and *Uilleann pipes* (0.063).

Q: *Which actor became M in the Bond film Skyfall?* A: *Ralph Fiennes* (0.651) and *Judi Dench* (0.348).

Q: *Which can last longer with out water a camel or a rat?* A: *A rat* (0.538) and *A camel* (0.461).

Q: *If Monday's child is fair of face what is Saturday's child?* A: *Work hard for a living* (0.093) and *Full of grace* (0.906).

Figure 2: Single-label queries with high epistemic uncertainty: Conditional normalized probability of the correct completion given repetitions of an incorrect response. Each figure shows the query and the considered two responses with their initial probabilities, as a response for the query, in parentheses (the first response is the correct one).

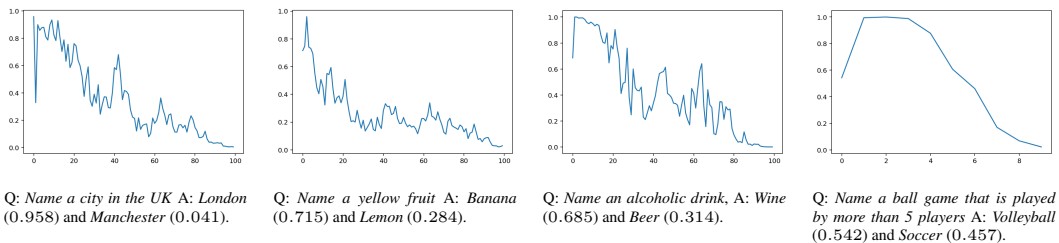

Q: *Name a city in the UK* A: *London* (0.958) and *Manchester* (0.041).

Q: *Name a yellow fruit* A: *Banana* (0.715) and *Lemon* (0.284).

Q: *Name an alcoholic drink,* A: *Wine* (0.685) and *Beer* (0.314).

Q: *Name a ball game that is played by more than 5 players* A: *Volleyball* (0.542) and *Soccer* (0.457).

Figure 3: Multi-label queries with aleatoric uncertainty: Conditional normalized probability of the first of the two provided responses, both of which are correct, given repetitions of the second response in the prompt. Each figure shows the query and the considered two responses with their initial probabilities, as a response for the query, in parentheses.

the correct response, *"London"*, reduces from approximately 1 to about 96% as we increase the number of repetitions of the incorrect response to 100. Figure 1 shows 3 more examples where, with initially low epistemic uncertainty in the response to the query (the aleatoric uncertainty is also low as we consider single-response queries), the correct response maintains a significant or non-negligible probability even in the presence of repetitions of incorrect information, while the probability of predicting the latter is increased.

Next, we consider a queries for which the model is more uncertain. For the prompt *"What is the national instrument of Ireland?"*, we observe that responses *"The harp"* and *"Uilleann pipes"* both have significant probabilities (the first answer is the correct one). This time, by incorporating the incorrect response in the prompt multiple times, the probability of the correct answer quickly collapses to near zero, as shown in Figure 2, with significant epistemic uncertainty.

Finally, we consider multi-label queries for which the LLM confidently knows a correct answer. This time, by incorporating a potential response in the prompt, the probabilities of other correct answers stay relatively large. Figure 3 shows four such examples.

# 4 Explanation through the lens of in-context vs. in-weight learning

The sensitivity of the response of an LLM to extra *in-context* information, as observed above, can already be observed in a single attention head as explained next.

We consider an idealized attention mechanism as follows. Let $\mathbf{Z} \in \mathbb{R}^{n \times d'}$ be an input matrix comprised of $n$ semantic feature vectors each of dimension $d'$. Each row is meant to represent a complete statement (such as *"What is the capital of the UK?"* or *"One answer to the question is Paris."*, etc.) rather than a single token. Let $X^\top \in \mathbb{R}^{1 \times d'}$ be the first row of $\mathbf{Z}$, which represents the *query* of interest, such as *"What is the capital of the UK?"*. Let $E^\top \in \mathbb{R}^{1 \times d'}$ be a special vector indicating the end of the input. The matrix $\mathbf{Z} \setminus X$, denoting the $\mathbf{Z}$ matrix without its first row, represents the *in-context* information.

We assume the ground-truth distribution $P$ is such that a query vector is mapped to its response, but a statement is simply copied. For example, for $V = $ *"What is the capital of the UK?"*, $P(\cdot \mid V)$ would be a distribution with support on *"London"* and its variations, while for $V' = $ *"What is the capital of the UK? One answer to the question is Paris."*, $P(\cdot \mid V')$ returns the same distribution. We assume a parameter matrix $\mathbf{W}^{\mathbf{V}}$ is learned such that $V^\top \mathbf{W}^{\mathbf{V}}$ estimates $P(\cdot \mid V)$ for vector $V$.

Let $\mathbf{W}^{\mathbf{Q}}, \mathbf{W}^{\mathbf{K}}, \mathbf{W}^{\mathbf{V}} \in \mathbb{R}^{d' \times d}$ be the query, key, and value matrices. A self-attention head with query $X$ and context $\mathbf{Z} \setminus X$ is defined as

$$f(\mathbf{Z}; \mathbf{W}^{\mathbf{Q}}, \mathbf{W}^{\mathbf{K}}, \mathbf{W}^{\mathbf{V}}) = \text{Softmax}\left(\frac{1}{\sqrt{d}} E^\top \mathbf{W}^{\mathbf{Q}}(\mathbf{Z}\mathbf{W}^{\mathbf{K}})^\top\right) \mathbf{Z}\mathbf{W}^{\mathbf{V}}$$

where the output of the softmax is a row vector of length $n$.

If $X$ has appeared many times in the training data, then parameters $\mathbf{W}^{\mathbf{Q}}$ and $\mathbf{W}^{\mathbf{K}}$ could be learned such that $E^\top \mathbf{W}^{\mathbf{Q}}(\mathbf{W}^{\mathbf{K}})^\top X$ is large, that is, $X$ is within the space spanned by the large principal components of the key-query matrix product. Then, no matter what in-context information appears in $\mathbf{Z}$, the probability assigned to $X$ will dominate the softmax, and we will have $\text{Softmax}\left(\frac{1}{\sqrt{d}} E^\top \mathbf{W}^{\mathbf{Q}}(\mathbf{Z}\mathbf{W}^{\mathbf{K}})^\top\right) \mathbf{Z} \approx X^\top$, and therefore $f(\mathbf{Z}; \mathbf{W}^{\mathbf{Q}}, \mathbf{W}^{\mathbf{K}}, \mathbf{W}^{\mathbf{V}}) \approx P(\cdot \mid X)$.

Now, consider the case that $X$ has not appeared many times in the training data, and vector $Y$ is copied in many rows of $\mathbf{Z}$. Then $E^\top \mathbf{W}^{\mathbf{Q}}(\mathbf{W}^{\mathbf{K}})^\top X$ could be small as $X$ is not in the span of the large principal components of the key-query matrix product. Therefore $f(\mathbf{Z}; \mathbf{W}^{\mathbf{Q}}, \mathbf{W}^{\mathbf{K}}, \mathbf{W}^{\mathbf{V}}) \approx Y$ since $\text{Softmax}\left(\frac{1}{\sqrt{d}} E^\top \mathbf{W}^{\mathbf{Q}}(\mathbf{Z}\mathbf{W}^{\mathbf{K}})^\top\right) \mathbf{Z} \approx Y^\top$. Even if $X$ is in the span, repeating $Y$ $t$ times in $\mathbf{Z}$ would give a $t$-times increased total weight to $Y$ inside the softmax, which can dominate the weight assigned to $X$ when $t$ is large enough, also resulting in $Y$ as the answer.

# 5 Metric of epistemic uncertainty and its estimation

In this section we apply iterative prompting to estimate the epistemic uncertainty of a language model about responding to some query. The idea is to utilize the different behavior patterns observed in Section 3, which can be used to differentiate between two modes of high uncertainty: when the aleatoric uncertainty is high vs. when only the epistemic uncertainty is high. We then apply our new uncertainty metric to design a score-based hallucination detection algorithm.

We will first present the uncertainty metric and its estimate for a distribution defined on the direct outputs of an LLM; the changes needed to take semantic equivalences of language into account are deferred to Appendix A (Kuhn et al., 2023; Farquhar et al., 2024).

Our uncertainty metric, similarly to the ones considered by, e.g., Wen et al. (2022); Osband et al. (2023), is based on analyzing the joint distribution of responses: if multiple responses are sampled jointly according to the ground-truth distribution, they should be independent (as one instantiation of a response should not affect other responses). To make this notion precise, we start with defining a notion of the joint distribution over responses given a query, derived from the language model through the prompting mechanism $\mathcal{F}$ defined in Section 2:

**Definition 5.1** (Pseudo joint distribution). Given a family of prompt functions $\mathcal{F}$, a conditional distribution $\mu \in \mathcal{P}$, and $n \in \mathbb{N}$, we use notation $\widetilde{\cdot}$ to denote a pseudo joint distribution defined as

$$\widetilde{\mu}(Y_1, \ldots, Y_n \mid x) = \mu(Y_1 \mid F_0(x))\,\mu(Y_2 \mid F_1(x, Y_1)) \cdots \mu(Y_n \mid F_{n-1}(x, Y_1, \ldots, Y_{n-1}))\,. \quad (1)$$

The above is a *pseudo* joint distribution since the standard conditioning in the chain-rule is replaced with prompt functions of the conditioning variables. In the following we focus on $\widetilde{Q}$ derived from the LLM and $\widetilde{P}$ derived from the ground truth.

**Remark 5.2** (Sampling from $\widetilde{Q}$). *Note that sampling from $\widetilde{Q}$ can be simply done through a chain-rule-like procedure as can be seen from the above definition, that is, to have $(Y_1, \ldots, Y_n) \sim \widetilde{Q}$ we draw $Y_1 \sim Q(\cdot \mid F_0(x))$, $Y_2 \sim Q(\cdot \mid F_1(x, Y_1))$, $Y_3 \sim Q(\cdot \mid F_2(x, Y_1, Y_2))$, and so on.*

In the rest of the paper we drop subscripts in joint distributions and conditioning on query $x$ (which is understood implicitly), for example, $\widetilde{P} \equiv \widetilde{P}_{Y_1 \cdots Y_n | x}$.

For any query $x \in \mathcal{X}$, let $\mathcal{Y}_{\widetilde{Q}}(x)$ denote the support of $\widetilde{Q}$. We make the following assumption about the ground truth, which states that the model $Q$ generates reasonable responses and that the distribution of such responses are independent according to the ground truth:

**Assumption 5.3** (Ground truth independence assumption). For any query $x \in \mathcal{X}$, *(i)* there exists a sequence of valid responses $\mathcal{Y}(x) \subset \mathcal{X}$ such that the ground-truth distribution satisfies

$$P(Y_t \mid F_{t-1}(x, Y_1, \ldots, Y_{t-1})) = P(Y_t \mid x) \qquad \text{for any } t \in \mathbb{N} \text{ and any } (Y_1, \ldots, Y_t) \in \mathcal{Y}(x);$$

*(ii)* $\mathcal{Y}_{\widetilde{Q}}(x) \subset \mathcal{Y}(x)$, that is, the model $Q$ generates reasonable responses.

Note that the above assumption is heavily dependent on our prompt construction and the assumption that $Y_1, \ldots, Y_{t-1}$ are valid responses; without these the independence assumption would not hold, for example, if $Y_1, \ldots, Y_t$ were partial answers, such as a step of an algorithm or a part of a story, or would completely redefine the problem (*"Disregard the previous question. Instead answer the following..."*), because in such cases $Y_t$ might indeed depend on the previous outputs $Y_1, \ldots, Y_{t-1}$. Roughly speaking, the assumption tells that the response distribution is insensitive to a query based on previously sampled responses. For example, for query $x =$*"Capital of the UK:"*, the probability of $Y_2 =$*"London"* essentially does not change if a city is $Y_1 =$*"Paris"*.

To measure epistemic uncertainty, we need to quantify how far the estimated pseudo joint distribution $\tilde{Q}$ is from the ground truth $\tilde{P}$. One natural choice is the following definition:

**Definition 5.4** (Epistemic uncertainty metric). Given an input $x \in \mathcal{X}$, we say that the epistemic uncertainty of $\widetilde{Q}$ is quantified by $D_{\mathrm{KL}}(\widetilde{Q}, \widetilde{P})$.

Here $D_{\mathrm{KL}}$ measures how well $\widetilde{Q}$ approximates $\widetilde{P}$ for a given query $x$. Namely, this metric determines if $\widetilde{Q}$ assigns a large probability to an event which has a small probability under $\widetilde{P}$. In case of LLMs, this means the LLM generates a sequence that is unlikely in the typical usage of the language. Given an input $x$, we want to estimate the above hallucination metric, but we only have access to $\widetilde{Q}$, and so computing it explicitly is impossible. However, next we show that under Assumption 5.3 we can *lower bound* $D_{\mathrm{KL}}(\widetilde{Q}, \widetilde{P})$ by a quantity which *only depends* on $\widetilde{Q}$ (the proof is given in Appendix E).

**Theorem 5.5.** *For all pseudo joint distributions $\widetilde{P}$ and $\widetilde{Q}$ satisfying Assumption 5.3, $D_{\mathrm{KL}}(\widetilde{Q}, \widetilde{P}) \geq I(\widetilde{Q})$.*

The lower bound in the theorem holds uniformly for all $\widetilde{P}$, and it is computable solely based on $\tilde{Q}$. This makes the bound applicable for decision making; in fact we chose to consider $D_{\mathrm{KL}}(\widetilde{Q}, \widetilde{P})$ as the measure of epistemic uncertainty (out of similar distance measures) since it admits this property.

Also, note that we have $I(\widetilde{Q}) = D_{\mathrm{KL}}(\widetilde{Q}, \widetilde{Q}^{\otimes})$, $\widetilde{Q}^{\otimes} = \prod_i \sum_{y \setminus i} \widetilde{Q}(y_1, \ldots, y_{i-1}, Y_i, y_{i+1}, \ldots, y_n)$. In general $\sum_{y \setminus i} \widetilde{Q}(y_1, \ldots, y_{i-1}, Y_i, y_{i+1}, \ldots, y_n) \neq \widetilde{Q}(Y_i)$, because the independence assumption Assumption 5.3 does not necessarily (and, in practice, almost never) holds for $Q$.

Finally, a quantity related to $D_{\mathrm{KL}}(\widetilde{Q}, \widetilde{P})$ is $D_{\mathrm{KL}}$ with arguments arranged in the opposite order, that is $D_{\mathrm{KL}}(\widetilde{P}, \widetilde{Q})$ which is a (query) conditional *excess risk* of the LLM-derived pseudo joint distribution $\widetilde{Q}$, under the logarithmic loss. Controlling the excess risk (for instance, upper-bounding it) for various algorithms is one of the central questions in learning theory, however it is a much harder task than the one we consider here, because for the former we need to theoretically control all sources of errors (such as generalization, estimation, and approximation error).

1: **Input**:

    $k \in \mathbb{N}$ ......................... sample size

4: Construct empirical distributions: let $Z = \sum_{j \in U} \mu(X_j)$, and for all $i \in U$, compute

## 5.1 A computable lower bound on epistemic uncertainty

Theorem 5.5 gives a lower bound on the epistemic uncertainty by the mutual information. However, to compute the mutual information term, in practice we need to evaluate $\widetilde{Q}$ on its entire support, which is potentially infinite. Practically speaking, it is impossible to observe probabilities of all strings under the language model and so we must rely on a finite sample. Therefore, we replace $\widetilde{Q}$ with an empirical distribution with a finite support; in the following we show that the error induced by such an approximation is controlled. To estimate the MI we employ the method given in Algorithm 1; for generality it is presented for an arbitrary (pseudo) joint distribution $\mu$, but we keep in mind that our case of interest is $\mu = \widetilde{Q}$. Note that most terms in the summations defining the product distribution $\widehat{\mu}^{\otimes}$ are zero (except the ones which correspond to the observed data). Adding $\gamma_1$ and $\gamma_2$ in the estimator $\widehat{I}_k(\gamma_1, \gamma_2)$ is intended to account for the total probability of missing observations, not included while constructing $\widehat{\mu}$ and $\widehat{\mu}^{\otimes}$, making sure the estimate is bounded.

The bias introduced by $(\gamma_1, \gamma_2)$ in the last equation allows us to rigorously bound the error in estimating $I(\mu)$ via $\widehat{I}_k(\gamma_1, \gamma_2)$, which is explored next. In particular, in Theorem 5.6 we prove a high-probability lower bound on $I(\mu)$ in terms of $\widehat{I}_k$. The core of controlling the estimation error is in accounting for the *missing mass*, or in other words, how much of $\mu$ we miss out by only observing a finite sample. In Appendix F, we present a more complete discussion and the proof of the bound on the estimation error for mutual information. Here we adapt this result to our particular case.

Define the missing mass as $U_k = \sum_{x \in \mathcal{X}^n} \mu(x) \, \mathbb{I}\big\{ x \notin \{X_1, \ldots, X_k\} \big\}$. Using this quantity, we are ready to present a non-asymptotic bound on the estimation error, which depends on the estimator $\widehat{I}_k(\gamma_1, \gamma_2)$, the expected missing mass, and the sample size:

**Theorem 5.6.** *Suppose that $\widehat{I}_k(\gamma_1, \gamma_2)$ is given by Algorithm 1, and assume that $\mathcal{X}$ is finite. For $\gamma_1 = 1/(k\,|\mathcal{X}^n|)$, and $\gamma_2$ satisfying $\gamma_2 \geq \gamma_1 + n(1 - Z)$, with probability at least $1 - \delta$, we have*

$$I(\mu) \geq (1 - \varepsilon_k) \, \widehat{I}_k(\gamma_1, \gamma_2) - \left( \frac{1}{k} + (1 + n \ln(1 + k\,|\mathcal{X}|)) \, \varepsilon_k \right) \quad \text{where} \quad \varepsilon_k = \mathbb{E}[U_k] + \sqrt{\frac{\ln(\frac{1}{\delta})}{k}} \, .$$

*Furthermore, given $\delta_{\mathrm{supp}} \in [0, 1)$, let $\tilde{\mathcal{X}} \subseteq \mathcal{X}^n$ such that $\mu(\tilde{\mathcal{X}}) \geq 1 - \delta_{\mathrm{supp}}$. Then, for $\gamma_1 = 1/(k\,|\tilde{\mathcal{X}}|)$, and $\gamma_2$ satisfying $\gamma_2 \geq \gamma_1 + n(1 - Z)$, with probability at least $1 - \delta$, we have*

$$I(\mu) \geq (1 - \varepsilon_k) \, \widehat{I}_k(\gamma_1, \gamma_2) - \left( \frac{1}{k} + (1 + \ln(1 + k\,|\tilde{\mathcal{X}}|)) \, (\delta_{\mathrm{supp}} + \varepsilon_k) \right) \, .$$

The theorem is a corollary of Theorem F.4 shown in Appendix F. Note that in Theorem 5.6 we consider two bounds. The first one is pessimistic in the sense that it does not expect that the samples carry much information about the support, and it is most suitable in situations where we expect $\mu$ to be spread out (uniformly) across its entire support. The price of not having samples covering the whole support in

this case is a factor $n \ln |\mathcal{X}|$ appearing in the bound. For example, in case of a language model with $10,000$ tokens, considering all possible strings of length $T$ tokens yields $n \ln |\mathcal{X}| = n \, T \ln(10000)$, and so $I(\mu) \geq (1 - \varepsilon_k) \, \widehat{I}_k(\gamma_1, \gamma_2) - (1/k + (1 + n \, T \ln(1 + k \, \ln(10000)))) \, \varepsilon_k)$. Arguably, in practice, such situations are rare, as in natural languages we will not encounter all possible strings. To this end, we consider an optimistic scenario where the *effective* support of $\mu$, denoted by $\tilde{\mathcal{X}}$, is small with high probability. In this case, we can replace the size of the support for strings of length $n$, $|\mathcal{X}|^n$, in the first bound with the effective support size $|\tilde{\mathcal{X}}|$, and we only pay essentially a factor $\ln(1 + k|\tilde{\mathcal{X}}|)$ instead of $n \ln(1 + k|\mathcal{X}|)$. In case the effective sample size is only polynomial in $n$, this leads to an exponential reduction in $n$ for the second term in the bounds. In fact, in Appendix F.4 we demonstrate some empirical evidence that on two question-answering benchmarks, $|\tilde{\mathcal{X}}|$ rarely exceeds $\approx 100$ with $\mu(\tilde{\mathcal{X}}) \geq 0.95$, while sampling responses from an LLM given a query.

Next we consider sufficient conditions for the estimator to converge to the mutual information. In particular, using the first bound in the theorem, we have (hiding logarithmic factors) $I(\mu) = \tilde{\Omega}((1 - \mathbb{E}[U_k]) \, \widehat{I}_k(\gamma_1, \gamma_2) - \mathbb{E}[U_k])$ as $k \to \infty$. This tells us that the rate of estimation error is essentially controlled by the expected missing mass $\mathbb{E}[U_k]$, which, as we will see, converges to zero as $k \to \infty$, however the decay can be very slow in general. For example, it is known that for a finite support of size $N$, $\mathbb{E}[U_k] \leq e^{-\frac{k}{N}}$ when $k \leq N$ and $\mathbb{E}[U_k] \leq N/(e \, k)$ otherwise (Berend and Kontorovich, 2012). For countable distributions with entropy bounded by $h$, one has $\mathbb{E}[U_k] \leq h/\ln(k)$.

Despite these pessimistic bounds, in reality we expect the expected missing mass to be significantly smaller, especially when $\mu$ is heavy-tailed. It is well-known that natural languages (and many artificial ones) follow a *Zipf* distribution, where probability of each word (or a text piece) is proportional to $1/\mathrm{freq}(\mathrm{text})^\alpha$ for some exponent $\alpha > 1$, where $\mathrm{freq}()$ is a frequency of occurrence in the corpus (Piantadosi, 2014). Then, we expect that $\mathbb{E}[U_k]$ should be much smaller than in such a case, since sampling from the tail of Zipf distribution is a rare event. To this end, in Appendix F we show that if $\tilde{Q}$ is Zipf with exponent $\alpha > 1$, then for any free parameter $\beta > 0$, $\mathbb{E}[U_k] = \mathcal{O}(k^{-(\frac{\alpha - 1}{\alpha} - \beta)})$. Hence, the rate at which the expected missing mass vanishes can be very fast (potentially matching a concentration rate $1/\sqrt{k}$ for $\alpha = 2$).

Finally in Appendix F.4 we present a data-dependent estimation of $\mathbb{E}[U_k]$ based on a concentration inequality for a missing mass and repetitive sampling from LLM, in the context of Q/A datasets showing that the expected missing mass is highly concentrated close to 0.

## 5.2 Score-based hallucination tests

Let $\widehat{I}_k(\gamma, x) \equiv \widehat{I}_k(\gamma)$ computed as in Algorithm 1 for $\mu = \tilde{Q}$, to emphasize the explicit dependence on the query $x$. The uncertainty estimate $\widehat{I}_k(\gamma, x)$ derived above can be used as a score indicating the strength of our belief that the LLM hallucinates for the given query $x$. Such a score can then be used to design *abstention* policies: if the response is deemed to be hallucinated, the system abstains from responding, while a response is provided otherwise. Score-based abstention methods usually compute a score chosen by the user (such as the response likelihood or the estimator $\widehat{I}(\gamma)$ discussed earlier), and declare hallucination if the score is above or below a threshold, which is determined through calibration. To detect hallucinations successfully, the threshold can be adjusted through *calibration* on a given task using a hold-out (ground-truth) sample, see, for instance, the paper of Yadkori et al. (2024) where this calibration is discussed in detail.

Given our estimated lower bound on the epistemic uncertainty, we can define an *abstention policy* (a policy which decides when the LLM should abstain from prediction) as $a_\lambda(x) = 0$ if $\widehat{I}_k(\gamma_1, \gamma_2, x) < \lambda$ and $a_\lambda(x) = 1$ if $\widehat{I}_k(\gamma_1, \gamma_2, x) \geq \lambda$, where $\lambda > 0$ is a threshold parameter tuned on a hold-out sample of some particular task. This policy abstains ($a_\lambda(x) = 1$) when the epistemic uncertainty in the prediction (response) is large. When the policy does not abstain ($a_\lambda(x) = 0$), any prediction from $\widehat{Q}$ can be served. In the experiments, we compare a number of scoring functions for detecting hallucinations, including $\widehat{I}(\gamma)$, the probability of the greedy (temperature zero) response, and an estimate of the entropy of the response distribution.

# 6   Experiments

In this section we evaluate our abstention method derived based on the MI estimate in Section 5.2 on a variety of closed-book open-domain question-answering tasks. In our experiments we either sweep through all abstention thresholds (Figure 4), or optimize the threshold on some calibration data, as explained in the description of the relevant experiment (Figure 5).

**Language model.**   We used a Gemini 1.0 Pro model (Gemini Team, Google, 2023) to generate outputs and scores. Similar results were obtained with a – much smaller – Gemini 1.0 Nano-1 model, which are deferred to Appendix H.

**Datasets.**   We consider three different datasets and their combinations: As base datasets, we consider *(i)* a random subset of $50,000$ datapoints from the TriviaQA dataset (Joshi et al., 2017), and *(ii)* the entire AmbigQA dataset (with 12038 datapoints) (Min et al., 2020). These datasets mostly contain single-label queries, and only contain a few multi-label ones.[5] Moreover, we created a multi-label dataset based on the WordNet dataset (Fellbaum, 1998): We extracted all (6015) datapoints from WordNet at depth $4$ or more of the `physical_entity` subtree. For each datapoint (`entity`, `children`) in WordNet, we constructed a query of the form *"Name a type of `entity`."* and `children` are considered target labels.

**Comparison of responses and computing the output distributions.**   We use the F1 score thresholded at $0.25$ to decide if two text sequences match. Additional details are provided in Appendix G.

**Baselines.**   We consider abstention policies based on four scoring methods: *(i)* the probability of the greedy response (denoted by $T0$); *(ii)* the semantic-entropy method of Kuhn et al. (2023) whose score is the entropy of $k = 10$ generated samples (denoted by S.E.); *(iii)* our proposed mutual information score as defined in Section 5 (and denoted by M.I.) with the choices of $k = 10$, $n = 2$, and $\gamma_1 = \gamma_2 = 0$ (the latter choice approximates the case that the number of potential responses can be very large in which case the theoretical choice of $\gamma_1$ and $\gamma_2$ would be very small); *(iv)* the self-verification method of Kadavath et al. (2022) (denoted by S.V.). Additional details are provided in Appendix G.

**Results.**   We consider the precision-recall (PR) trade-off for the various methods on the different datasets. Here, *recall* is the percentage of queries where the method does not abstain, and *precision* is the percentage of correct decisions among these queries.[6]  Figure 4ab show PR-curves for the baselines and the proposed method on TriviaQA and AmbigQA. As can be seen, our method is better than the $T0$ and S.V. baselines, but performs similarly to the S.E. method. This is because the TriviaQA and AmbigQA datasets contain mostly single-label queries, and therefore a first-order method such as S.E. is sufficient to detect hallucinations. The AmbigQA dataset contains a few multi-label queries, but upon closer inspection, we observe that the LLM has low entropy on most of these queries.Therefore, a first-order method can perform as well as our method on such queries. Our proposed method, as well as the baselines, make no mistakes on the WordNet dataset (as the prediction of the LLM is always correct), hence we omit those results. The S.V. baseline performs significantly worse than the other methods when the recall is not high (is below about 0.8).

The similar performance for the S.E. and M.I. methods shown in Figure 4ab is due to the fact that the LLM has low entropy on most multi-label queries. However, ideally, an LLM should have higher entropy on multi-label queries (which would demonstrate broader knowledge, not focusing on a single possible answer). To include such queries, we mix the TriviaQA and AmbigQA datasets with our WordNet-based dataset with "truely" multi-label queries as constructed above. To enhance the intended effect, we filter our WordNet dataset by keeping only queries with entropy higher than $0.7$ (approximately the entropy of the uniform distribution over two atoms). Then we have $842$ remaining datapoints in WordNet. Note that when considered in isolation, both our proposed method and the semantic entropy method rarely make mistakes on this dataset. Then we create two new datasets by combining our $842$ WordNet datapoints with $842$ randomly selected datapoints from TriviaQA and AmbigQA, respectively, resulting in the TriviaQA+WordNet and AmbigQA+WordNet datasets. Figure 4cd show PR-curves for the S.E. and M.I. methods on these two combined datasets. Apart from low recall values, the performance of the S.E. method degrades noticeably with the addition

---

[5]Note that the multi-label queries in these datasets typically behave as single-label ones in the sense that the LLM assigns overwhelming probability to a dominant response.

[6]In some figures, for better illustration, we show the *error rate* which is one minus the precision.

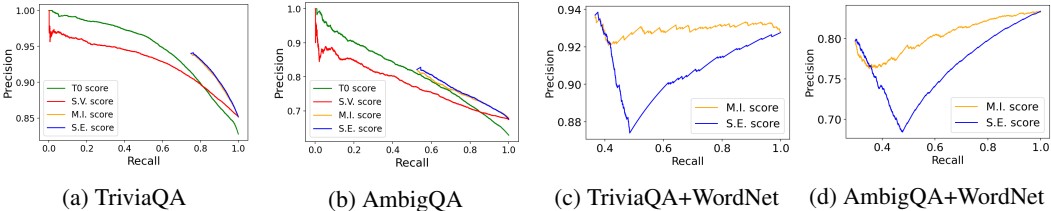

(a) TriviaQA  (b) AmbigQA  (c) TriviaQA+WordNet  (d) AmbigQA+WordNet

Figure 4: PR-curve for the baseline and the proposed methods on various datasets. On the TriviaQA and AmbigQA datasets, M.I. and S.E. perform nearly identically, but they outperform the $T0$ and S.V. baselines. For the S.E. and M.I. methods, the responses for a large number of queries can be clustered into a single group, and therefore the semantic entropy and mutual information scores are zero. This is why the starting point of their curves is at a higher recall values. On the TriviaQA+WordNet and AmbigQA+WordNet datasets with a significant number of high entropy multi-label queries, M.I. outperforms the S.E. baseline. The methods perform nearly identical on the not shown recall area.

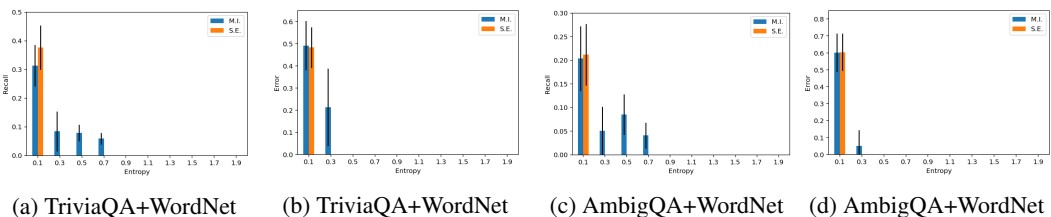

(a) TriviaQA+WordNet  (b) TriviaQA+WordNet  (c) AmbigQA+WordNet  (d) AmbigQA+WordNet

Figure 5: Recall and error rates (one minus precision: percentage of mistakes when not abstaining) of the proposed and the baseline method on TriviaQA+WordNet and AmbigQA+WordNet datasets. On TriviaQA+WordNet and AmbigQA+WordNet datasets, the methods are calibrated at target loss of $0.05$ and $0.15$, respectively. On the x-axis, the queries are partitioned according to the entropy of the LLM's output. Error bars show 2 standard deviation confidence intervals (based on 10 repetitions). While the first-order S.E. method has similar recall and error rates to those of the proposed M.E. method on low-entropy queries, its recall values are nearly zero for queries with higher entropy.

of extra multi-label data. This precision/recall curve might look somewhat strange (with precision sometimes increasing with recall); this is due to the fact that both methods are always correct on the large number of high-entropy WordNet queries, where the LLM's default predictions are correct.

The hardness with the combined datasets is that the predominantly single-label datasets (TriviaQA, AmbigQA) might need a different calibration threshold than the multi-label WordNet dataset, and this is better handled by our proposed method than by S.E. To better illustrate the improved abstention properties of our method, we examine how the two methods handle when the output of the LLM is diverse (i.e., has high entropy). In order to do this, we perform the following experiment: We create a calibration dataset by adding 500 random datapoints from the WordNet dataset to 500 random datapoints from TriviaQA, and another such random dataset for test. We determine the abstention thresholds on the calibration dataset for both the S.E. and the M.E. methods,[7] and measure the performance (error rate, i.e., 1 minus precision, and recall) of the resulting abstention policies on the test set. We repeat this process 10 times and report mean values and 95% confidence intervals with Gaussian approximation. We perform a similar evaluation process for mixtures of AmbigQA and WordNet datasets. Figure 5 show that while the S.E. method has similar recall and error rates to those of the proposed method on low-entropy queries, its recall values are much lower for queries with higher entropy, while the M.E. method makes only few mistakes on these queries.

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

# A  Additional algorithms for taking semantic equivalences into account

Although Theorem 5.5 and Theorem 5.6 provide a lower bound for the divergence between the output distribution $Q$ of the LLM and the ground truth $P$, these distributions ignore the semantic equivalences between texts, and hence they are less concentrated than their variants which only consider semantically different outputs. Given a semantic equivalence definition, similarly to Kuhn et al. (2023); Farquhar et al. (2024), we propose constructing new ground-truth distribution $P'$ and LLM output distribution $Q'$, where we cluster together semantically equivalent texts, and the probability of a cluster is the sum of probabilities of all semantically equivalent texts in that cluster. We use a similarity function $s$ to define semantic equivalences: two texts are considered equivalent if their similarity is greater than a given threshold $\tau$. Our choices for similarity functions in the experiments are described in Section 6. We assume the similarity function and the similarity threshold induce a clustering of the space $\mathcal{X}$, i.e., $s(Y, Y') \geq \tau$ for $Y, Y' \in \mathcal{X}$ if and only if they are in the same cluster.

In practice, rather than constructing the aforementioned distribution $Q'$ explicitly, we can draw samples from $Q'$ by sampling from $Q$ and aggregating samples according to their clusters. The modified uncertainty-estimation algorithm is given in Algorithm 2 in Appendix A. The estimator is constructed using only (semantically) different elements in the sample (the indices of these representative elements are collected in $S$), that is, we do not account for duplicate samples and we aggregate probabilities of samples that are lexically different but semantically equivalent. Algorithm 2 works with the *aggregated* probability distribution $\mu' = \widetilde{Q'}$ (line 4) by summing over cumulative probabilities over clusters. Note that $D_{\mathrm{KL}}(\mu) \geq D_{\mathrm{KL}}(\mu')$ by monotonicity property of KL-divergence (Polyanskiy and Wu, 2024, Theorem 2.16) (this is because $\mu'$ is defined on a smaller support). Therefore, Theorem 5.6 implicitly gives a bound on $I(\mu')$, and eventually we have $I(\mu) \geq I(\mu')$. More importantly, we can also directly apply Theorem 5.5 and Theorem 5.6 to the distributions $P'$ and $Q'$ and obtain that $D_{\mathrm{KL}}(\widetilde{Q'}, \widetilde{P'}) \geq I(\widetilde{Q'})$, giving a lower bound on the much reduced (epistemic) uncertainty after taking semantic equivalences into account.

1: **Input**:

    $\mu \in \mathcal{M}_1(\mathcal{X}^n)$ .................... any (pseudo-) joint distribution over $\mathcal{X}^n$

    $k \in \mathbb{N}$ .......................... sample size

    $\gamma_1, \gamma_2 \geq 0$ ...................... stabilization parameters (typically selected as $1/k$)

    $s : \mathcal{X}^n \times \mathcal{X}^n \to \mathbb{R}$ .............. a similarity function

    $\tau \in \mathbb{R}$ .......................... a similarity threshold

2: Independently sample tuples $X_1, \ldots, X_k \sim \mu \in \mathcal{M}_1(\mathcal{X}^n)$

3: Construct a set of indices of unique elements $U = \left\{ i \in [k] \ : \ X_i \neq X_j \quad \forall j < i \right\}$

4: Construct cluster centers $S \subset U$ according to the similarity function such that for all $i, t \in S$, we have $s(X_i, X_t) < \tau$ and the cluster associated with $X_i$ is $D(i) = \left\{ j \in U : s(X_i, X_j) \geq \tau \right\}$. Compute the aggregated probabilities: for all $i \in S$,

$$\mu'(X_i) = \sum_{j \in D(i)} \mu(X_j)$$

5: Construct empirical distributions: for all $i \in S$,

$$\widehat{\mu}(X_i) = \frac{\mu'(X_i)}{Z}, \quad \text{where} \quad Z = \sum_{j \in S} \mu'(X_j)$$

$$\widehat{\mu}^{\otimes}(X_i) = \prod_{j=1}^{n} \sum_{t \in S : X_{t,j} = X_{i,j}} \hat{\mu}(X_{t,1}, \ldots, X_{i,j}, \ldots, X_{t,n})$$

6: Compute estimate

$$\widehat{I}_k(\gamma_1, \gamma_2) = \sum_{i \in S} \widehat{\mu}(X_i) \ln \left( \frac{\widehat{\mu}(X_i) + \gamma_1}{\widehat{\mu}^{\otimes}(X_i) + \gamma_2} \right)$$

Algorithm 2: MI estimator. Python implementation with usage example is given in Appendix B.

1: **Input**:

$\mu \in \mathcal{M}_1(\mathcal{X})$ ..................... any distribution over $\mathcal{X}$

$k \in \mathbb{N}$ ......................... sample size

$\gamma_1, \gamma_2 \geq 0$ ..................... stabilization parameters (typically selected as $1/k$)

$s : \mathcal{X} \times \mathcal{X} \to \mathbb{R}$ ................. a similarity function

$\tau \in \mathbb{R}$ ........................ a similarity threshold

2: Independently sample outputs $X_1, \ldots, X_k \sim \mu \in \mathcal{M}_1(\mathcal{X})$

3: Construct a set of indices of unique elements $U = \big\{i \in [k] \; : \; X_i \neq X_j \quad \forall j < i\big\}$

4: Construct cluster centers $S \subset U$ according to the similarity function: for all $i, t \in S$, we have $s(X_i, X_t) < \tau$ and cluster associated with $X_i$ is $D(i) = \big\{j \in U \; : \; s(X_i, X_j) \geq \tau\big\}$. Aggregated probabilities: for all $i, t \in S$,

$$\mu'_1(X_i) = \sum_{j \in D(i)} \mu(X_j), \qquad \mu'_2(X_t \,|\, X_i) = \sum_{j \in D(t)} \mu(X_j \,|\, X_i)$$

5: Construct empirical distributions: for all $i, t \in S$,

$$\widehat{\mu}_1(X_i) = \frac{\mu'_1(X_i)}{Z}, \quad \text{where} \quad Z = \sum_{j \in S} \mu'_1(X_j)$$

$$\widehat{\mu}_2(X_t \,|\, X_i) = \frac{\mu'_2(X_t \,|\, X_i)}{Z_i}, \quad \text{where} \quad Z_i = \sum_{j \in S} \mu'_2(X_j \,|\, X_i)$$

$$\widehat{\mu}(X_i, X_t) = \widehat{\mu}_1(X_i)\widehat{\mu}_2(X_t \,|\, X_i), \qquad \widehat{\mu}^{\otimes}(X_i, X_t) = \widehat{\mu}_1(X_i) \sum_{j \in S} \widehat{\mu}_1(X_j)\widehat{\mu}_2(X_t \,|\, X_j)$$

6: Compute estimate

$$\widehat{I}_k(\gamma_1, \gamma_2) = \sum_{i, t \in S} \widehat{\mu}(X_i, X_t) \ln\left(\frac{\widehat{\mu}(X_i, X_t) + \gamma_1}{\widehat{\mu}^{\otimes}(X_i, X_t) + \gamma_2}\right)$$

Algorithm 3: Alternative MI estimator. A usage example is given in Appendix C

## B Implementation and usage examples of Algorithm 1 and Algorithm 2

In this section we present an implementation of Algorithm 1 and Algorithm 2 in Python with a simple usage example. In particular, the code given in Listing 1 generates a synthetic joint distribution over binary tuples with correlated elements (function `create_synthetic_distribution`). Then, we compute an exact mutual information of the distribution (function `compute_MI_exactly`) and use implementation of our estimator (function `MI_estimator`) to estimate a mutual information. This is done for various sample sizes, number of random variables, and levels of correlation (a single experiment is implemented by `run_experiment`) The results of these multiple experiments are eventually presented in as plots showing convergence of the estimate to the exact value of the mutual information. In practical applications, synthetic joint distribution (function `create_synthetic_distribution`) can be replaced by an LLM-derived pseudo-joint distribution (see Definition 5.1). More detailed description of each function is given in Appendix B.1.

The example can be easily copied from `listing.tex` within https://arxiv.org/src/2406.02543.

Listing 1: Implementation and usage examples of Algorithm 1 and Algorithm 2 on a synthetic joint distribution

```python
# Copyright 2024 DeepMind Technologies Limited.
#
# Licensed under the Apache License, Version 2.0 (the "License");
# you may not use this file except in compliance with the License.
# You may obtain a copy of the License at
#
# https://www.apache.org/licenses/LICENSE-2.0
#
# Unless required by applicable law or agreed to in writing, software
# distributed under the License is distributed on an "AS IS" BASIS,
# WITHOUT WARRANTIES OR CONDITIONS OF ANY KIND, either express or implied.
# See the License for the specific language governing permissions and
# limitations under the License.
#
from itertools import product, combinations
import numpy as np
from matplotlib import pyplot as plt

def create_synthetic_distribution(space, temp):
  potential = lambda z: np.mean([x * y for x, y in combinations(z, 2)])
  y = np.array([-np.exp(potential(x) / temp) for x in space])
  return y / y.sum()

def sample_from_joint_distribution(space, joint_dist, k):
  indices = np.arange(len(joint_dist))
  sampled_indices = np.random.choice(indices, p=joint_dist, size=k)
  sampled_tuples = space[sampled_indices]
  return sampled_tuples, sampled_indices

def cluster(tuples, joint_dist):
  return tuples, joint_dist

def sample_from_joint_distribution_and_cluster(space, joint_dist, k):
  sampled_tuples, sampled_tuple_indices = sample_from_joint_distribution(space, joint_dist, k)
  _, indices_of_uniques_in_sample = np.unique(sampled_tuples, axis=0, return_index=True)
  sampled_tuples = sampled_tuples[indices_of_uniques_in_sample]
  sampled_tuple_indices = sampled_tuple_indices[indices_of_uniques_in_sample]
  joint_dist_on_sample = joint_dist[sampled_tuple_indices]
  sampled_tuples, joint_dist_on_sample = cluster(sampled_tuples, joint_dist_on_sample)
  return joint_dist_on_sample, sampled_tuples

def compute_MI_exactly(space, mu):
  total = 0
  for (x_i, x) in enumerate(space):
    mu_x = mu[x_i]
    mu_x_prod = 1
    for i in range(len(x)):
      marg_indices = [j for (j, z) in enumerate(space) if z[i] == x[i]]
      mu_x_prod *= mu[marg_indices].sum()
    total += mu_x * np.log(mu_x/mu_x_prod)

  return total

def MI_estimator(sampled_tuples, mu_on_sample, gamma_1, gamma_2):
  """Implements MI estimator (Algorithm 1).
```

```python
    Args:
        sampled_tuples: A numpy array of tuples sampled from the distribution after deduplication and clustering.
        mu_on_sample: A numpy array of probabilities of the clusters.
        gamma_1: stabilization parameter.
        gamma_2: stabilization parameter.

    Returns: (float) mutual information.
    """

    # Constructing empirical distribution (\hat{\mu})
    hat_mu_on_sample = mu_on_sample / mu_on_sample.sum()

    # Constructing empirical product distribution (\hat{\mu}^{\otimes})
    hat_mu_prod_on_sample = np.zeros((len(hat_mu_on_sample),))
    for (x_i, x) in enumerate(sampled_tuples):
        hat_mu_x_prod = 1
        for i in range(len(x)):
            marg_indices = [j for (j, z) in enumerate(sampled_tuples) if z[i] == x[i]]
            hat_mu_x_prod *= hat_mu_on_sample[marg_indices].sum()

        hat_mu_prod_on_sample[x_i] = hat_mu_x_prod

    # Computing MI estimate
    mi_estimate = hat_mu_on_sample * np.log((hat_mu_on_sample + gamma_1) / (hat_mu_prod_on_sample + gamma_2) )
    mi_estimate = mi_estimate.sum()
    return mi_estimate

def run_experiment(n, temp, ax):
    np.random.seed(1)
    space = np.array(list(product([-1, 1], repeat=n)))
    mu = create_synthetic_distribution(space, temp=temp)
    mi_exact = compute_MI_exactly(space, mu)
    k_range = np.linspace(10, 1000, 20, dtype=int)

    all_mi_estimate = []
    for k in k_range:
        mu_on_sample, sampled_tuples = sample_from_joint_distribution_and_cluster(space=space, joint_dist=mu, k=k)

        gamma_1 = gamma_2 = 1/k
        mi_estimate = MI_estimator(sampled_tuples, mu_on_sample, gamma_1, gamma_2)
        all_mi_estimate.append(mi_estimate)

    ax.axhline(mi_exact, linewidth=3, label="MI (exact value)", color="black")
    ax.plot(k_range, all_mi_estimate, linewidth=3, label="MI estimator")
    ax.grid(); ax.legend(); ax.set_xlabel("k"); ax.set_ylabel("MI estimate"); ax.set_title(r"$n=$"+str(n)+r", $\tau=$"+str(temp))

temp_range = [0.01, 0.1, 1, 10]
n_range = [2, 4, 8]

fig, axs = plt.subplots(len(temp_range), len(n_range), figsize=(5*len(temp_range), 5*len(n_range)), squeeze=False)
fig.suptitle(r"""MI estimation of $n$-dimensional distribution $\propto \exp(-\sum_{i < j}^n x_i x_j / \tau)$""")

for (i, temp) in enumerate(temp_range):
    for (j, n) in enumerate(n_range):
        ax = axs[i,j]
        run_experiment(n=n, temp=temp, ax=ax)
plt.subplots_adjust(wspace=0.4, hspace=0.4)
plt.show()
```

## B.1   Additional documentation for functions in Listing 1

- def create_synthetic_distribution(space, temp)

    Creates synthetic distribution which introduces dependencies between variables.

    Args:
        space: a list of tuples that the joint distribution is supported on (e.g. a cartesian product).
        temp: temperature parameter.

    Returns:
        A numpy array of probabilities (same length as space).

- def sample_from_joint_distribution(space, joint_dist, k)

    Samples k tuples from a joint distribution.

    Args:

space: a list of tuples that the joint distribution is supported on (e.g. a cartesian product).
joint_dist: probability distribution (1-D numpy array) where each entry is a probability of a tuple.
k: sample size.

Returns:
    A numpy array of tuples sampled from the distribution.
    A numpy array of indices of sampled tuples in space.

- def cluster(tuples, joint_dist)

  Clusters tuples and aggregates probabilities of them in the same cluster.

  Args:
      tuples: A numpy array of tuples sampled from the distribution
      after deduplication.
      joint_dist: probability distribution (1-D numpy array) where each entry
      is a probability of a tuple.

  Returns:
      A numpy array of tuples sampled from the distribution each represening
      a cluster.
      A numpy array of probabilities of clusters. Each probability is the
      aggregate of probabilities of all tuples in the cluster.

- def sample_from_joint_distribution_and_cluster(space, joint_dist, k)

  Samples k tuples from a joint distribution and retains only
  representative elements (removes all duplicates).

  Args:
      space: a list of tuples that the joint distribution is supported on (e.g. a cartesian product).
      joint_dist: probability distribution (1-D numpy array) where each entry is a probability of a tuple.
      k: sample size.

  Returns:
      A numpy array of tuples sampled from the distribution after deduplication.
      A numpy array of probabilities of deduplicated tuples.

- def compute_MI_exactly(space, mu)

  Computes mutual information of probability distribution mu exactly
  Args:
      space: Tuple space (cartesian product).
      mu: probability distribution (1-D numpy array).

  Returns: (float) mutual information.

- def MI_estimator(sampled_tuples, mu_on_sample, gamma_1, gamma_2)

  Implements MI estimator (Algorithm 1).

  Args:
      sampled_tuples: A numpy array of tuples sampled from the distribution after deduplication and clustering.
      mu_on_sample: A numpy array of probabilities of the clusters.
      gamma_1: stabilization parameter.
      gamma_2: stabilization parameter.

  Returns: (float) mutual information.

- def run_experiment(n, temp, ax)

  Runs one experiment comparing exact mutual information estimation with
  Algorithm 1.  Plots results.

  Args:
      n: number of variables in a joint distribution.
      temp: temperature of a Gibbs distribution (joint distribution). Higher
      temperature typically means smaller MI.
      ax: pyplot axis object for plotting.

# C   Usage example of Algorithm 3

Algorithm 3 is a slight modification of Algorithm 2 that we use in our experiments. We first explain Algorithm 3 via an example and then highlight the differences with Algorithm 2. In order to explain the implementation, we consider a running example with query $x =$ *"What is the capital of the UK?"*, F1 score as the similarity function $s$, similarity threshold $\tau = 0.25$, and number of samples $k = 5$. Algorithm 3 also takes the LLM distribution $Q$ as input $\mu = Q$.

Given the query, in step (2) of Algorithm 3, we sample $k$ outputs from $Q$. Let's assume these samples are $X_1 =$ *"London"*, $X_2 =$ *"London"*, $X_3 =$ *"London, UK"*, $X_4 =$ *"Paris"*, and $X_5 =$ *"Berlin"*. In step (3), we construct a set of indices of unique elements. In our example, we would have $U = \{1, 3, 4, 5\}$. In step (4), we cluster responses and aggregate probabilities of each cluster. More precisely, if the F1 score of two responses is above 0.25, then they are in the same cluster. In our example, we have that $\text{F1}(X_1, X_3) > 0.666 > 0.25$ and $\text{F1}(X_1, X_4) = \text{F1}(X_1, X_5) = 0$, and therefore cluster centers are $S = \{1, 4, 5\}$. For query $x$, let's assume LLM probabilities are

$$Q(X_1 \,|\, x) = 0.5, \quad Q(X_3 \,|\, x) = 0.2, \quad Q(X_4 \,|\, x) = 0.1, \quad Q(X_5 \,|\, x) = 0.05, \quad \cdots$$

Also assume conditional distributions are

$$Q(X_1 \,|\, F_1(x, X_1)) = 0.6, \, Q(X_3 \,|\, F_1(x, X_1)) = 0.15, \, Q(X_4 \,|\, F_1(x, X_1)) = 0.05,$$
$$Q(X_5 \,|\, F_1(x, X_1)) = 0.04, \cdots$$

and so on (we have omitted writing $Q(. \,|\, F_1(x, X_4))$ and $Q(. \,|\, F_1(x, X_5))$). Then after step (4), the aggregated probabilities are

$$Q'(X_1 \,|\, x) = 0.7, \quad Q'(X_4 \,|\, x) = 0.1, \quad Q'(X_5 \,|\, x) = 0.05, \quad \cdots$$

and aggregated conditional probabilities are

$$Q'(X_1 \,|\, F_1(x, X_1)) = 0.75, \quad Q'(X_4 \,|\, F_1(x, X_1)) = 0.05, \quad Q'(X_5 \,|\, F_1(x, X_1)) = 0.04, \cdots$$

and so on (we have similar aggregations for $Q'(. \,|\, F_1(x, X_4))$ and $Q'(. \,|\, F_1(x, X_5))$). Next, in step (5), we construct empirical estimates. We will have that $Z = 0.85$ and

$$\widehat{Q}_1(X_1 \,|\, x) \approx 0.82, \quad \widehat{Q}_1(X_4 \,|\, x) \approx 0.11, \quad \widehat{Q}_1(X_5 \,|\, x) \approx 0.05, \quad \cdots$$

For estimated conditional distributions, we will have $Z_1 = 0.84$, and

$$\widehat{Q}_2(X_1 \,|\, F_1(x, X_1)) \approx 0.89, \quad \widehat{Q}_2(X_4 \,|\, F_1(x, X_1)) \approx 0.06, \quad \widehat{Q}_2(X_5 \,|\, F_1(x, X_1)) = 0.04, \cdots$$

and so on. The joint distribution $\widehat{Q}(., . \,|\, x)$, the product of marginals $\widehat{Q}^{\otimes}(., . \,|\, x)$, and the estimated mutual information $\widehat{I}_k$ are trivially obtained by the equations in steps (5) and (6).

Next, we highlight the differences between Algorithm 2 (with the choice of $n = 2$) and Algorithm 3. The input distribution to Algorithm 2 is the pseudo-joint distribution $\widetilde{Q}$, while the input to Algorithm 3 is the LLM distribution $Q$. So in step (2) of Algorithm 2, each sample is a tuple such as (*"London"*, *"Paris"*), while a sample in step (2) of Algorithm 3 is an LLM output such as *"London"*. Steps (3) and (4) of Algorithm 2 are similarly modified, and now the similarity function is defined over tuples.

# D Related work

In this section we present an overview of the related literature.

## D.1 Bayesian neural networks

In a Bayesian framework, we can estimate the epistemic uncertainty by the uncertainty in the posterior distribution (Neal, 2012; Gal, 2016; Wang and Holmes, 2024). Implementing a Bayesian neural network however can be very challenging.

## D.2 Iterative prompting

A number of iterative prompting strategies are developed to improve the factuality of LLMs (Chen et al., 2023; Krishna, 2023; Laban et al., 2024). The idea is to follow-up the LLM response with another question such as *"Are you sure?"*. Krishna et al. (2024) show that such strategies might in fact degrade LLM truthfulness due to a pattern of apologetic responses. Krishna et al. (2024) propose improved iterative prompting strategies, where instead of asking the LLM to re-think its response, the same question is posed again. They also propose strategies to collect more supporting facts and refine the final response accordingly. Li et al. (2024) propose an iterative prompting that instructs LLM to generate justifications for each answer before evaluating the correctness of the final answer. Different from these works, we assess hallucinations by measuring how LLM response changes with our iterative prompting scheme.

Perhaps the most related work to ours is the parallel and independent work of Ahdritz et al. (2024). Similar to us, Ahdritz et al. (2024) observe that in presence of high epistemic uncertainty, an LLM is more likely to copy the information provided in its context. For a given query, Ahdritz et al. (2024) propose considering top-$k$ completions of the model, and then computing the entropy of the model conditioned on an iterative prompt composed of the original query and each completion. The minimum of these entropies is considered as a measure of the epistemic uncertainty. The method as it is, might fail on single-response queries where the model has low uncertainty. Nevertheless, we can design a two-stage process where we only consider completions that have probability higher than certain threshold in the first stage, and then compute the entropy of the model conditioned on an iterative prompt composed of the original query and each candidate completion in the second stage. By a proper tuning of the threshold of the first stage, we can potentially avoid mis-classification of low-uncertainty single-response queries. Tuning this threshold, however, would introduce extra complications. In contrast, we propose a principled test using a mutual information score that is guaranteed to be a lower bound on the KL-divergence between the LLM and the ground-truth. Further, we provide a mechanistic explanations for why LLMs behave as described in the presence of high epistemic uncertainty.

## D.3 Training models with pairs of responses

Wen et al. (2022); Osband et al. (2023); Johnson et al. (2024) show that we can decouple epistemic and aleatoric uncertainty if we train a model with paired observations.

We discuss the more recent work of Johnson et al. (2024) in more detail. The proposed approach first estimates a model $\hat{p}_{Y_1,Y_2|x}(y_1, y_2|x)$ over pairs using a training dataset of the form "query, first observation, second observation". At test time, for a prompt $x$ and response $y$, Johnson et al. (2024) consider

$$\hat{V}(y \mid x) = \hat{p}_{Y_1}(y \mid x) \left( \hat{p}_{Y_2|Y_1}(y \mid y, x) - \hat{p}_{Y_1}(y \mid x) \right)$$
$$= \hat{p}_{Y_1,Y_2}(y_1, y_2 \mid x) - \hat{p}_{Y_1}(y \mid x)\hat{p}_{Y_1}(y \mid x)$$

as a measure of epistemic uncertainty. Assume that an equivalence class $\Phi$ (that maps a prompt to the set of equivalent prompts) is given, and let $\nu(. \mid \Phi(x))$ be a distribution (say, uniform) over class $\Phi(x)$. If the trained model is second order calibrated with respect to the equivalence class and the

distribution $\nu$, i.e.

$$\hat{p}_{Y_1}(y_1 \mid x) = \sum_{x' \in \Phi(x)} \nu(x' \mid \Phi(x)) p(y_1 \mid x') \,,$$

$$\hat{p}_{Y_1, Y_2}(y_1, y_2 \mid x) = \sum_{x' \in \Phi(x)} \nu(x' \mid \Phi(x)) p(y_1 \mid x') p(y_2 \mid x') \,,$$

then it follows from definitions that in the class associated with $x$,

$$\sum_{x' \in \Phi(x)} \nu(x' \mid \Phi(x)) (\hat{p}_{Y_1}(y \mid x') - p(y \mid x'))^2 = \sum_{x' \in \Phi(x)} \nu(x' \mid \Phi(x)) \hat{V}(y \mid x') \,.$$

The quantity on the right-hand side is a measure of epistemic uncertainty. Notice that the equality states a coverage result, and it is not point-wise. Requiring the model to be second order calibrated is also a strong condition and ensuring it is highly non-trivial.

## D.4 Epistemic neural nets

*Ensemble methods* are based on the classical idea of bootstrap for confidence estimation (Tibshirani and Efron, 1993), where multiple estimators for the regression function, each computed on a perturbed version of the data (e.g., by drawing samples from the empirical distribution over the data), are combined.

The empirical distribution of the resulting estimates is then used to construct confidence intervals. While many of these methods can be interpreted as sample-based approximations to Bayesian methods, model-hyperparameter selection (e.g., scale of perturbations, learning) for ensemble methods is typically done using a validation on holdout data (a subset of the training data). Many recent papers have studied ensemble methods in the context of deep learning and reinforcement learning (Osband et al., 2016; Lakshminarayanan et al., 2017a; Malinin and Gales, 2020). In the context of LLMs, the methods require training multiple language models, which is very expensive. Osband et al. (2023) introduces epistemic neural networks (epinets), which approximate ensemble methods by training a single network with an artificially injected (controlled) source of randomness. Rabanser et al. (2022) proposes to use intermediate model checkpoints to quantify the uncertainty of the final model in its responses. While these approaches aim to mimic the bootstrap procedure during prediction, their validity is not justified by theoretical considerations, and hence remain heuristic approximations.

## D.5 Hallucination detection using first-order methods

First-order methods consider variance in the response distribution as a measure of hallucination (Kadavath et al., 2022; Cole et al., 2023; Manakul et al., 2023; Lin et al., 2023; Kuhn et al., 2023; Wang et al., 2022; Jiang et al., 2024; Zhang et al., 2023; Zhao et al., 2024; Yadkori et al., 2024). A common limitation of these approaches is that they are only applicable to prompts where there exists a *single* correct response, as they aim for detecting if one response (or multiple responses with the same meaning) is dominant. On the other hand, when multiple responses are correct, there is an *aleatoric uncertainty* in the ground truth: If an LLM *correctly* assigns non-negligible scores to multiple correct responses, most of these (if not all) will be declared as hallucination since, by design, only very few (typically at most one) responses can have scores higher than the threshold at the same time. Thus, hallucination detectors unaware of aleatoric uncertainty will invalidate most of the correct answers.

Yona et al. (2024) design a method that generates multiple responses, and then aggregates them into a single response at a (typically higher) granularity level where no further uncertainty (contradiction) is left compared to the generated responses. Although not a strictly first order method, it does not differentiate between aleatoric and epistemic uncertainty.

### D.5.1 Asking language models to quantify uncertainty (self-verification)

Kadavath et al. (2022) propose to use LLM self-prompting to measure a model's uncertainty in its responses. More specifically, for a given query, a number of responses are generated, and then the model is queried if the responses are correct. For this query, the log-probability of "True" is returned as a measure of uncertainty. Related approaches are studied by Mielke et al. (2022).

## D.6 Uncertainty estimation based on sensitivity to contexts

Kassner and Schütze (2020); Zhao et al. (2021) show that an LLM's responses can be influenced by irrelevant contexts. Longpre et al. (2021); Neeman et al. (2022) study two sources of knowledge: parametric knowledge stored in the network weights, and contextual knowledge retrieved from external sources. They view reliance of the model on its parametric knowledge and ignoring relevant contextual information as hallucination. These works are mainly motivated by situations where the LLM's knowledge is outdated and it is instructed to use the (new) contextual information. Accordingly, they design strategies to prioritize contextual information over parametric knowledge. Longpre et al. (2021) also show that larger models are more likely to ignore in-context information in favor of in-weight information. They propose creating training data with modified contextual information so that the model learns to favor the contextual information. Neeman et al. (2022) propose to train a model that predicts two answers: one based on parametric knowledge and one based on contextual information.

Similarly to Neeman et al. (2022), Li et al. (2023) aims to design a mechanism such that the model's behavior is influenced more by relevant context than by its parametric knowledge (controllability), while the model is robust to irrelevant contexts (robustness). They improve controllability and robustness using finetuning.

Hou et al. (2024) study an approach to estimate model uncertainty due to ambiguity in a question. For a given question, their method generates multiple input clarification questions, and a new question is formed by augmenting the original question with each clarification question. The clarification questions are generated using an LLM with the aim of removing ambiguity in the question. This is different than the problem we study as the model can be uncertain about the answer even if the query itself has no ambiguity. For such queries, the method of Hou et al. (2024) might decide that no clarification is needed, and therefore there is no uncertainty.

## D.7 Hallucination detection using internal states of LLMs

There are a number of papers that try to extract knowledge/truthfulness by inspecting hidden-layer activations of LLMs (Burns et al., 2023; Azaria and Mitchell, 2023; Chen et al., 2024a,b; Yin et al., 2024). Such methods clearly require access to the LLM's internal states, which is not always possible, and severely limits the applicability of these methods.

# E   Omitted proofs

*Proof of Theorem 5.5.* In the following we will use abbreviations

$$\sum_y = \sum_{y_1,\ldots,y_n} , \qquad \sum_{y\setminus i} = \sum_{y_1,\ldots,y_{i-1},y_{i+1},\ldots,y_n}$$

where each $n$-tuple $y$ belongs to $\mathcal{Y}$. Now,

$$D_{\mathrm{KL}}(\widetilde{Q},\widetilde{P}) = -H(\widetilde{Q}) + \sum_y \widetilde{Q}(y_1,\ldots,y_n) \ln \frac{1}{\widetilde{P}(y_1,\ldots,y_n)}$$

$$= -H(\widetilde{Q}) + \sum_y \widetilde{Q}(y_1,\ldots,y_n) \ln \frac{1}{\prod_i P\big(y_i \mid F_{i-1}(y_1,\ldots,y_{i-1})\big)}$$

$$\text{(using Definition 5.1)}$$

$$= -H(\widetilde{Q}) + \sum_y \widetilde{Q}(y_1,\ldots,y_n) \ln \frac{1}{\prod_i P(y_i)} . \quad \text{(by the independence assumption)}$$

Focusing on the last (cross-entropy) term

$$\sum_y \widetilde{Q}(y_1,\ldots,y_n) \ln \frac{1}{\prod_i P(y_i)}$$

$$= \sum_y \widetilde{Q}(y_1,\ldots,y_n) \sum_i \ln \frac{1}{P(y_i)}$$

$$= \sum_i \sum_{y_i} \sum_{y\setminus i} \widetilde{Q}(y_1,\ldots,y_n) \ln \frac{1}{P(y_i)}$$

$$\overset{(a)}{\geq} \sum_i \sum_{y_i} \sum_{y\setminus i} \widetilde{Q}(y_1,\ldots,y_n) \ln \frac{1}{\sum_{y\setminus i} \widetilde{Q}(y_1,\ldots,y_n)}$$

$$= \sum_y \widetilde{Q}(y_1,\ldots,y_n) \ln \frac{1}{\prod_i \sum_{y\setminus i} \widetilde{Q}(y_1,\ldots,y_n)}$$

where in $(a)$ we used the fact that entropy is no larger than cross-entropy. Thus,

$$D_{\mathrm{KL}}(\widetilde{Q},\widetilde{P}) \geq \sum_y \widetilde{Q}(y_1,\ldots,y_n) \ln \frac{\widetilde{Q}(y_1,\ldots,y_n)}{\prod_i \sum_{y\setminus i} \widetilde{Q}(y_1,\ldots,y_n)} = I(\widetilde{Q}; Y_1,\ldots,Y_n) .$$

$$\square$$

## F Estimation of mutual information and missing mass problem

In this section, we discuss how to estimate the mutual information from a finite sample, which may not cover the full distribution. To control the estimation error, we first introduce the concept of *missing mass*.

### F.1 The missing mass problem

Let $\mathcal{X}$ be a countable set and suppose that $X_1, \ldots, X_k \sim \mu \in \mathcal{M}_1(\mathcal{X}^n)$ independently. In the following $x$ is used as an element of $\mathcal{X}^n$ rather than the query (as in Section 5). Then, the missing mass is defined as the random variable

$$U_k = \sum_{x \in \mathcal{X}^n} \mu(x)\, \xi(x)\,, \qquad \xi(x) = \mathbb{I}\{x \notin \{X_1, \ldots, X_k\}\}\,.$$

Here we are primarily interested in two questions: *(i)* how quickly $U_k$ approaches the expected missing mass $\mathbb{E}U_k$, where it is not hard to see that

$$\mathbb{E}U_k = \sum_{x \in \mathcal{X}^n} \mu(x)(1 - \mu(x))^k\,;$$

and *(ii)* we are also interested in giving an estimate for $\mathbb{E}U_k$ given $\mu$ and $k$. The first question is answered by the following theorem:

**Theorem F.1** (Concentration of a missing mass (Berend and Kontorovich, 2013))**.** *For any $t > 0$, we have an upper-tail bound*

$$\mathbb{P}\left(U_k > \mathbb{E}U_k + t\right) \leq e^{-tk^2}\,,$$

*and moreover for a universal constant $c \approx 7.6821$, we have an lower-tail bound*

$$\mathbb{P}\left(U_k < \mathbb{E}U_k - t\right) \leq e^{-ctk^2}\,.$$

Notably $U_k$ exhibits a sub-gaussian concentration (i.e. $1/\sqrt{k}$), which is surprisingly fast. As we will see next, the main bulk of the error incurred for missing a subset of the support is hidden in $\mathbb{E}U_k$.

In particular, when $\mathcal{X}$ is finite with $|\mathcal{X}| = N$, Berend and Kontorovich (2012) showed that

$$\mathbb{E}U_k \leq \begin{cases} e^{-\frac{n}{N}}, & \text{if } n \leq N; \\ \frac{N}{e\,n}, & \text{if } n > N. \end{cases}$$

In the countably infinite $\mathcal{X}$, we cannot generally have a non-trivial bound on $\mathbb{E}U_k$ only in terms of $n$. In fact, Berend and Kontorovich (2012) show a bound that depends on $\mu$ which is expected to be finite for rapidly decaying atoms. Interestingly, when the entropy of $\mu$ is bounded, one has the following result (Berend et al., 2017):

**Theorem F.2.** *Let $H(\mu) \leq h < \infty$. For all $n \geq 1$, we have $\mathbb{E}U_k \leq \frac{h}{\sum_{i=1}^{k} i^{-1}} \leq \frac{h}{\ln(n)}$.*

Note that these estimates are very pessimistic, and in reality we expect the expected missing mass to be significantly smaller. Since natural (and many artificial) languages follow a Zipf distribution (Piantadosi, 2014), we expect that $\mathbb{E}[U_k]$ should be much smaller than in the above cases, since sampling from the tail of a Zipf distribution is a rare event. In Appendix F.4 we show the following:

**Corollary F.3** (Expected missing mass of Zipf distribution)**.** *Consider distribution $\mu(i) = i^{-\alpha}/H(\alpha, N)$ for $i \in [N]$, where $\alpha > 1$ and $H(\alpha, N) = \sum_{i=1}^{N} i^{-\alpha}$. Then, for any $\beta > 0$,*

$$\mathbb{E}[U_k] = \mathcal{O}\left(k^{-\left(\frac{\alpha-1}{\alpha} - \beta\right)}\right)\,.$$

*Proof.* The statement followss by combining Lemma F.9 and Proposition F.10. □

## F.2 Estimating mutual information from the partial support

Our goal is to estimate

$$I(\mu) = D_{\mathrm{KL}}(\mu, \mu^{\otimes}) = \sum_{x \in \mathcal{X}^n} \mu(x) \ln \left( \frac{\mu(x)}{\mu^{\otimes}(x)} \right)$$

by only having access to $X_1, \ldots, X_k \sim \mu$. Note that that the sample might cover only some part of the support of $\mathcal{X}$ and therefore we are facing a missing mass problem. In the following we consider estimator $\widehat{I}_k(\gamma)$ given by Algorithm 1.

In particular in Appendix F.3 we show the following

**Theorem F.4.** *Fix $\tilde{\mathcal{X}} \subseteq \mathcal{X}^n$. Fix $\gamma_1 > 0$ and suppose that $\gamma_2 \geq n(1 - Z) + \gamma_1$. Then for any fixed $\delta \in (0, 1)$, with probability at least $1 - \delta$,*

$$(1 - \varepsilon_k)\, \widehat{I}_k(\gamma_1, \gamma_2) - \left( |\tilde{\mathcal{X}}|\gamma_1 + \ln \left( e + \frac{e}{\gamma_1} \right) \left( \mu(\mathcal{X}^n \setminus \tilde{\mathcal{X}}) + \varepsilon_k \right) \right) \leq I(\mu)$$

*where*

$$\varepsilon_k = \mathbb{E}U_k + \sqrt{\frac{\ln(\frac{1}{\delta})}{k}}\,.$$

In particular, Theorem F.4 implies the following:

**Corollary F.5.** *Under conditions of Theorem F.4, there exists $(\gamma_1^*, \gamma_2^*) \in (0, 1)^2$ such that*

$$(1 - \varepsilon_k)\, \widehat{I}_k(\gamma_1^*, \gamma_2^*) - \left( \frac{1}{k} + (1 + n \ln \left( 1 + k\,|\mathcal{X}| \right)) \varepsilon_k \right) \leq I(\mu)\,.$$

Note that, choosing any of the upper bounds on $\mathbb{E}U_k$ discussed in Appendix F.1, we can see that Corollary F.5 implies asymptotic convergence in as a sense

$$\lim_{k \to \infty} \widehat{I}_k(\gamma_1^*, \gamma_2^*) \leq I(\mu)\,.$$

## F.3 Proof of Theorem F.4

The proof will heavily rely on the simple fact that

$$1 - \xi(x) = \begin{cases} 1, & \text{if } x \in \{X_1, \ldots, X_k\}; \\ 0, & \text{otherwise.} \end{cases} \tag{2}$$

Recalling that $S = \left\{ i \in [k] \ : \ X_i \neq X_j \quad \forall j < i \right\}$, this immediately implies the following connection between $U_k$ and the quantities used in Algorithm 1:

**Proposition F.6.** *We have that*

$$\sum_{j \in S} \mu(X_j) = \sum_{x \in \mathcal{X}^n} (1 - \xi(x))\, \mu(x) = 1 - U_k\,.$$

Recall that the product distribution of $\mu$ is defined as

$$\mu^{\otimes}(x) = \prod_{i=1}^{n} \sum_{x \backslash i} \mu(x_1, \ldots, x_{i-1}, x_i, x_{i+1}, \ldots, x_n)\,.$$

Note that we use $\sum_{x\setminus i} \mu(\cdots)$ instead of $\mu(x_i)$ since these are not necessarily equal for some $\mu$. Now, using the definitions of $\widehat{I}_k, \widehat{\mu}$, and $\widehat{\mu}^{\otimes}$,

$$\widehat{I}_k(\gamma_1, \gamma_2) = \frac{1}{Z} \sum_{i \in S} \mu(X_i) \left( \ln\left( \frac{\mu(X_i)}{Z} + \gamma_1 \right) - \ln\left( \widehat{\mu}^{\otimes}(X_i) + \gamma_2 \right) \right)$$

$$= \frac{1}{Z} \sum_{x \in \mathcal{X}^n} (1 - \xi(x))\, \mu(x) \left( \ln\left( \frac{\mu(x)}{Z} + \gamma_1 \right) - \ln\left( \widehat{\mu}^{\otimes}(X_i) + \gamma_2 \right) \right) \qquad \text{(by Eq. (2))}$$

$$= \frac{1}{Z} \sum_{x \in \mathcal{X}^n} (1 - \xi(x))\, \mu(x) \left( \ln\left( \frac{\mu(x) + \gamma_1}{\mu^{\otimes}(x) + \gamma_1} \right) + \ln\left( \frac{\mu^{\otimes}(x) + \gamma_1}{\widehat{\mu}^{\otimes}(x) + \gamma_2} \right) + \ln\frac{1}{Z} \right)$$

$$= \underbrace{\frac{1}{Z} \sum_{x \in \mathcal{X}^n} \mu(x) \ln\left( \frac{\mu(x) + \gamma_1}{\mu^{\otimes}(x) + \gamma_1} \right)}_{(i)} + \underbrace{\frac{1}{Z} \sum_{x \in \mathcal{X}^n} \xi(x)\, \mu(x) \ln\left( \frac{\mu^{\otimes}(x) + \gamma_1}{\mu(x) + \gamma_1} \right)}_{(ii)} + \underbrace{\ln\frac{1}{Z}}_{(iii)}$$

$$+ \underbrace{\frac{1}{Z} \sum_{x \in \mathcal{X}^n} (1 - \xi(x))\, \mu(x) \ln\left( \frac{\mu^{\otimes}(x) + \gamma_1}{\widehat{\mu}^{\otimes}(x) + \gamma_2} \right)}_{(iv)}$$

Now we control each of the terms individually. To control $(i)$ we will first need the fact that $q \ln((q + \gamma_1)/p) \le q \ln(q/p) + \gamma_1$ for any $q, p \in [0, 1], \gamma_1 > 0$. Note that this follows since

$$q \ln\left( \frac{q + \gamma_1}{p} \right) = q \ln\left( 1 + \frac{\gamma_1}{q} \right) + q \ln\left( \frac{q}{p} \right) \le \gamma_1 + q \ln\left( \frac{q}{p} \right) \qquad (3)$$

using that $\ln(1 + a) \le a$ for $a > -1$. Getting back to $(i)$, and using the aforementioned inequality, we get

$$(i) = \frac{1}{Z} \sum_{x \in \mathcal{X}^n} \mu(x) \ln\left( \frac{\mu(x) + \gamma_1}{\mu^{\otimes}(x) + \gamma_1} \right)$$

$$= \frac{1}{Z} \sum_{x \in \tilde{\mathcal{X}}} \mu(x) \ln\left( \frac{\mu(x) + \gamma_1}{\mu^{\otimes}(x) + \gamma_1} \right) + \frac{1}{Z} \sum_{x \in \mathcal{X}^n \setminus \tilde{\mathcal{X}}} \mu(x) \ln\left( \frac{\mu(x) + \gamma_1}{\mu^{\otimes}(x) + \gamma_1} \right)$$

$$\le \frac{1}{Z} \sum_{x \in \tilde{\mathcal{X}}} \mu(x) \ln\left( \frac{\mu(x) + \gamma_1}{\mu^{\otimes}(x) + \gamma_1} \right) + \frac{1}{Z} \ln\left( \frac{1 + \gamma_1}{\gamma_1} \right) \mu(\mathcal{X}^n \setminus \tilde{\mathcal{X}})$$

$$\le \frac{1}{Z} \sum_{x \in \tilde{\mathcal{X}}} \left( \mu(x) \ln\left( \frac{\mu(x)}{\mu^{\otimes}(x)} \right) + \gamma_1 \right) + \frac{1}{Z} \ln\left( 1 + \frac{1}{\gamma_1} \right) \mu(\mathcal{X}^n \setminus \tilde{\mathcal{X}}) \qquad \text{(by Equation (3))}$$

$$= \frac{1}{Z} \left( D_{\mathrm{KL}}(\mu, \mu^{\otimes}) + |\tilde{\mathcal{X}}|\, \gamma_1 \right) + \frac{1}{Z} \ln\left( 1 + \frac{1}{\gamma_1} \right) \mu(\mathcal{X}^n \setminus \tilde{\mathcal{X}}).$$

Furthermore,

$$(ii) \le \frac{1}{Z} \sum_{x \in \mathcal{X}^n} \xi(x)\, \mu(x) \ln\left( 1 + \frac{1}{\gamma_1} \right) = \frac{1 - Z}{Z} \ln\left( 1 + \frac{1}{\gamma_1} \right).$$

Next, observe that $(iii) \le \ln(1/Z)$. Finally, term $(iv)$ is controlled through the following fact shown at the end of this section:

**Lemma F.7.** *Suppose that $\gamma_1 \ge 0$ while $\gamma_2 \ge \gamma_1 + n(1 - Z)$. Then,*

$$\frac{1}{Z} \sum_{x \in \mathcal{X}^n} (1 - \xi(x))\mu(x) \ln\left( \frac{\mu^{\otimes}(x) + \gamma_1}{\widehat{\mu}^{\otimes}(x) + \gamma_2} \right) \le 0.$$

Putting everything together, we obtain

$$\widehat{I}_k(\gamma_1, \gamma_2) \le \frac{1}{Z} \left( D_{\mathrm{KL}}(\mu, \mu^{\otimes}) + |\tilde{\mathcal{X}}|\gamma_1 \right) + \frac{1}{Z} \ln\left( 1 + \frac{1}{\gamma_1} \right) \left( \mu(\mathcal{X}^n \setminus \tilde{\mathcal{X}}) + 1 - Z \right) + \ln(1/Z).$$

Finally, multiplying through by $Z$ the entire inequality, and using the fact that $Z \ln(1/Z) \le 1 - Z$, we get

$$Z \widehat{I}_k(\gamma_1, \gamma_2) \le D_{\mathrm{KL}}(\mu, \mu^\otimes) + |\tilde{\mathcal{X}}| \gamma_1 + \ln\left(1 + \frac{1}{\gamma_1}\right)\left(\mu(\mathcal{X}^n \setminus \tilde{\mathcal{X}}) + 1 - Z\right) + 1 - Z$$

$$\le D_{\mathrm{KL}}(\mu, \mu^\otimes) + |\tilde{\mathcal{X}}| \gamma_1 + \ln\left(e + \frac{e}{\gamma_1}\right)\left(\mu(\mathcal{X}^n \setminus \tilde{\mathcal{X}}) + 1 - Z\right).$$

To complete the proof we need to give a lower bound on $Z$. Note that $Z = 1 - U_k$ by the definition of $Z$ and Proposition F.6, and so by Theorem F.1

$$\mathbb{P}\left(1 - \mathbb{E}U_k > 1 - U_k + t\right) \le e^{-tk^2}.$$

Using this concentration bound together with the choices of $\gamma$ (also setting $\delta_{\mathrm{supp}} = 0$ for the first inequality in the main statement) completes the proof of Theorem F.4. $\qquad\square$

*Proof of Lemma F.7.* Observe that

$$\widehat{\mu}^\otimes(x) = (1 - \xi(x)) \prod_{j=1}^n \sum_{t \in S: X_{t,j} = x_j} \hat{\mu}(X_{t,1}, \ldots, x_j, \ldots, X_{t,n})$$

$$= \frac{1}{Z^n}(1 - \xi(x)) \prod_{j=1}^n \sum_{t \in S: X_{t,j} = x_j} \mu(X_{t,1}, \ldots, x_j, \ldots, X_{t,n})$$

$$= \frac{1}{Z^n}(1 - \xi(x)) \prod_{j=1}^n \sum_{x' \in \mathcal{X}^n} (1 - \xi(x')) \mathbb{I}\{x'_j = x_j\} \mu(x'_1, \ldots, x_j, \ldots, x'_n)$$

$$= \frac{1}{Z^n}(1 - \xi(x)) \prod_{j=1}^n \sum_{x'^{\setminus j}} (1 - \xi(x'_1, \ldots, x_j, \ldots, x'_n)) \mu(x'_1, \ldots, x_j, \ldots, x'_n).$$

Now, using that fact that

$$\sum_{x'^{\setminus j}} \xi(x'_1, \ldots, x_j, \ldots, x'_n) \mu(x'_1, \ldots, x_j, \ldots, x'_n)$$

$$\le \sum_{x'^{\setminus j}, x_j} \xi(x'_1, \ldots, x_j, \ldots, x'_n) \mu(x'_1, \ldots, x_j, \ldots, x'_n)$$

$$= 1 - Z$$

we arrive at

$$\widehat{\mu}^\otimes(x) \ge \frac{1}{Z^n}(1 - \xi(x))\left(\prod_{j=1}^n \left(\sum_{x'^{\setminus j}} \mu(x'_1, \ldots, x_j, \ldots, x'_n) + Z - 1\right)\right)_+$$

$$\overset{(a)}{\ge} \frac{1}{Z^n}(1 - \xi(x))\left(\prod_{j=1}^n \sum_{x'^{\setminus j}} \mu(x'_1, \ldots, x_j, \ldots, x'_n) - n(1 - Z)\right)_+$$

$$= \frac{1}{Z^n}(1 - \xi(x))\left(\mu^\otimes(x) - n(1 - Z)\right)_+$$

where to get $(a)$ we used:

**Proposition F.8.** *For any $p_1, \ldots, p_n \in [0, 1]$ and $a \ge 0$, we have*

$$\prod_{i=1}^n (p_i - a) \ge \left(\prod_{i=1}^n p_i\right) - n\,a.$$

*Proof.* The statement following by lower-bounding the left-hand side by its linearization in $a$ (derivative at 0), while realizing that it is a convex function of $a$. $\qquad\square$

The above gives us that

$$\frac{1}{Z} \sum_{x \in \mathcal{X}^n} (1 - \xi(x))\mu(x) \ln \left( \frac{\mu^{\otimes}(x) + \gamma_1}{\widehat{\mu}^{\otimes}(x) + \gamma_2} \right)$$

$$\leq \frac{1}{Z} \sum_{x \in \mathcal{X}^n} (1 - \xi(x))\mu(x) \ln \left( \frac{\mu^{\otimes}(x) + \gamma_1}{\frac{1}{Z^n} (1 - \xi(x)) (\mu^{\otimes}(x) - n(1 - Z))_+ + \gamma_2} \right)$$

and focusing on the case $1 - \xi(x) = 1$ (otherwise both sides are 0) the above equals to

$$\frac{1}{Z} \sum_{x \in \mathcal{X}^n} \mu(x) \ln \left( \frac{\mu^{\otimes}(x) + \gamma_1}{\frac{1}{Z^n} (\mu^{\otimes}(x) - n(1 - Z))_+ + \gamma_2} \right)$$

$$\leq \frac{1}{Z} \sum_{x \in \mathcal{X}^n} \mu(x) \ln \left( \frac{\mu^{\otimes}(x) + \gamma_1}{(\mu^{\otimes}(x) - n(1 - Z))_+ + \gamma_2} \right)$$

$$\leq \frac{1}{Z} \sum_{x \in \mathcal{X}^n} \mu(x) \ln \left( \frac{\mu^{\otimes}(x) + \gamma_1}{\mu^{\otimes}(x) - n(1 - Z) + \gamma_2} \right)$$

$$\leq 0$$

by setting $\gamma_2 \geq \gamma_1 + n(1 - Z)$. $\qquad\square$

### F.4  Expected missing mass under Zipf distribution

We will rely on some machinery used by Ohannessian and Dahleh (2010) who established distribution-dependent bounds on the expected missing mass. As before let $\mu$ be supported on a countable set. The *accrual function* is defined as

$$F(v) = \sum_{\mu(i) \leq v} \mu(i) \qquad (v \in [0, 1])$$

and moreover the *accrual rates* are defined as

$$\underline{\rho} = \liminf_{v \to 0} \frac{\ln F(v)}{\ln v} , \qquad \overline{\rho} = \limsup_{v \to 0} \frac{\ln F(v)}{\ln v}$$

We use the following result:

**Lemma F.9** (Ohannessian and Dahleh, 2010, Theorem 1). *Let $\mu$ have lower and upper accrual rates $0 < \underline{\rho} \leq \overline{\rho} < \infty$. Then for every $\beta > 0$ there exists $k_0$ such that for all $k > k_0$ we have:*

$$k^{-(\overline{\rho}+\beta)} \leq \mathbb{E}[U_k] \leq k^{-(\underline{\rho}-\beta)}$$

*or, equivalently, for every $\beta > 0$ we have that $\mathbb{E}[U_k]$ is both $\Omega(k^{-(\overline{\rho}+\beta)})$ and $\mathcal{O}(k^{-(\underline{\rho}-\beta)})$.*

**Proposition F.10.** *Consider the distribution $\mu(v) = i^{-\alpha}/H(\alpha, N)$ for $i \in [N]$ where $\alpha > 1$ and $H(\alpha, N) = \sum_{i=1}^{N} i^{-\alpha}$. Then, $\underline{\rho} = \Omega(\frac{\alpha-1}{\alpha})$ as $N \to \infty$.*

*Proof.* The idea is to use Lemma F.9 to give an upper bound on the missing mass. Therefore, we need to establish a lower bound on $\ln F(v)$. For now, abbreviate

$$u = (v \, H(\alpha, N))^{-\frac{1}{\alpha}} .$$

First note that for some $1 \leq u \leq N$

$$\sum_{i \geq u}^{N} i^{-\alpha} \geq \int_{u}^{N} (1 + i)^{-\alpha} \, \mathrm{d}i = \frac{1}{\alpha - 1} \left( (1 + u)^{1-\alpha} - (1 + N)^{1-\alpha} \right) .$$

On the other hand,

$$\sum_{i=1}^{N} i^{-\alpha} \leq \int_{1}^{N} (1+i)^{-\alpha}\, \mathrm{d}i \leq \frac{1}{\alpha-1}\left(1 - N^{1-\alpha}\right).$$

So,

$$\ln F(v) \geq \ln\left((1+u)^{1-\alpha} - (1+N)^{1-\alpha}\right) - \ln(1 - N^{1-\alpha})$$
$$\geq \ln\left((1+u)^{1-\alpha} - (1+N)^{1-\alpha}\right)$$

and then

$$\ln F(v) = \Omega\left((1-\alpha)\ln(1+u)\right) \qquad (\text{as } N \to \infty)$$
$$= \Omega\left((1-\alpha)\ln(u)\right)$$
$$= \Omega\left((1-\alpha)\ln((v\,H(\alpha,N))^{-\frac{1}{\alpha}})\right)$$
$$= \Omega\left(\frac{\alpha-1}{\alpha}\,\ln(v) + \frac{\alpha-1}{\alpha}\ln H(\alpha,N)\right)$$
$$= \Omega\left(\frac{\alpha-1}{\alpha}\,\ln(v)\right).$$

$\square$

**Data-dependent estimate of the expected missing mass** We perform an experiment designed to give a data-dependent estimate of the expected missing mass $\mathbb{E}[U_k]$ for some specific datasets. Clearly, we cannot simply apply a concentration bound discussed in Appendix F.1 since the complete support of the pseudo joint distribution derived from the LLM is unknown. To this end, we approximate it with a finite support driven by the language model itself. In particular, given a query we sample responses (at temperature 0.9) until their total probability mass reaches 0.95 or we reach 1000 responses per query. In case of TriviaQA, we performed 1233 queries in total. The mean and the median number of unique responses per query was eventually 118.3 and 22, respectively. In case of the AmbigQA dataset, we performed 700 queries, while the mean and the median number of unique responses was 277 and 69, respectively.

At this point, we denote the set of responses by $\tilde{\mathcal{X}}$ and let $\tilde{U}_k$ be the missing mass computed on $\tilde{\mathcal{X}}$. Then, we have

$$\mathbb{E}[U_k] \leq U_k + \sqrt{\frac{\ln(\frac{1}{\delta})}{k}} \leq \tilde{U}_k + U_k - \tilde{U}_k + \sqrt{\frac{\ln(\frac{1}{\delta})}{k}} \leq \tilde{U}_k + 1 - P(\tilde{\mathcal{X}}) + \sqrt{\frac{\ln(\frac{1}{\delta})}{k}},$$

which can be computed in practice. In Figure 6 we present our results in the form of empirical distributions of different quantities, where each observation corresponds to a single query. We compute the bounds for TriviaQA and AmbigQA datasets (see Section 6 for details about these datasets). From Figure 6 we can conclude that the expected missing mass for both datasets is very small: Both the missing mass computed on $\tilde{\mathcal{X}}$ and the resulting upper bound on $\mathbb{E}[U_k]$ are concentrated close to 0, while the cumulative probability of the approximate support $\tilde{\mathcal{X}}$ is close to 1 most of the time, showing that our approximations are meaningful.

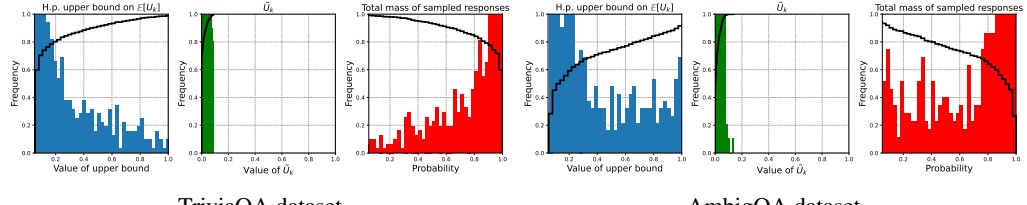

TriviaQA dataset                                         AmbigQA dataset

Figure 6: Distributions of bounds on the missing mass. The left figure for each dataset presents the empirical distribution of the upper bounds on the missing mass $\mathbb{E}[U_k]$. The middle figure presents the empirical distribution of $\tilde{U}_k$, the missing mass computed on a finite support approximation (where the support is obtained by taking samples from the LLM until a cumulative probability of 95% or 1000 samples are achieved). The right graph shows the empirical distribution of $P(\tilde{\mathcal{X}})$, the cumulative probabilities of all responses generated by the language model. For each figure, one observation (sample) corresponds to a single query. The black curves represent the corresponding empirical cumulative distribution functions for the upper bounds on $\mathbb{E}U_k$ and for $\tilde{U}_k$, and the empirical survival function (1 minus the empirical distribution function) for the distribution of $P(\tilde{\mathcal{X}})$.

# G   Additional experiments details

**Comparison of responses and computing the output distributions.** We use the F1 score[8] thresholded at $0.25$ to decide if two text sequences match. When multiple responses are sampled, we approximate the output distribution of an LLM in a semantically meaningful way by collapsing matching responses into a single response: we sample $k = 10$ responses at temperature $0.9$ for each query, and after eliminating repetitions, all those that match (according to the F1 score) are considered identical and their probabilities are aggregated. We only consider queries for which the greedy (temperature zero) and at least one of the random responses are shorter than 20 characters. This is because the F1 score (as a match function) and log-probabilities (as a measure of uncertainty) are less reliable for longer sequences. After this filtering, we are left with 38870 datapoints for TriviaQA, 5315 datapoints for AmbigQA, and 3296 datapoints for WordNet.

As shown in prior works (e.g. Kuhn et al. (2023); Yadkori et al. (2024)), we can use LLM self-prompting to obtain more reliable text comparisons specially for longer outputs. Such an approach however is computationally much more expensive.

**Baselines.** We consider abstention policies based on four scoring methods. The first three are as follows: *(i)* the probability of the greedy response (denoted by $T0$); *(ii)* the semantic-entropy method of Kuhn et al. (2023) whose score is the entropy of $k = 10$ generated samples (denoted by S.E.). To calculate entropy, we first aggregate probabilities of equivalent responses and normalize the probabilities so that they sum to 1 (as described above); and *(iii)* our proposed mutual information score as defined in Section 5 (and denoted by M.I.) with the choices of $k = 10$, $n = 2$, and $\gamma_1 = \gamma_2 = 0$ (the latter choice approximates the case that the number of potential responses can be very large in which case the theoretical choice of $\gamma_1$ and $\gamma_2$ would be very small). To calculate the mutual information, as shown in Algorithm 3 (given in Appendix A), we first generate $k = 10$ random samples. Then for any response $Y$, we calculate the probability of all generated responses given the prompt $F_1(x, Y)$. We construct estimates $\widehat{Q}(Y)$ and $\widehat{Q}(Y'|Y)$ by aggregating probabilities of equivalent responses, and normalizing the probabilities so that they sum to 1.

The calculation of the mutual information is slightly different than the algorithms presented in Algorithm 1 and Algorithm 2 and takes advantage of the available log-likelihood function in LLMs. Notice that the input $\mu$ in Algorithm 3 is the LLM's output distribution $Q$ as opposed to being the pseudo joint distribution $\widetilde{Q}$ in Algorithm 1. Another difference is that the similarity function $s$ now takes two texts as input (as opposed to taking two $n$-dimensional arrays of texts as inputs in Algorithm 2). As explained earlier, we use the F1 score as the similarity function and we use $\tau = 0.25$ as the similarity threshold.

Each baseline also has a default choice which is taken when the relevant score is above a threshold, and hence the method does not abstain. For $T0$, the default choice is the greedy (temperature zero) response. For S.E., the default choice is the response with the highest (aggregate) probability among the generated random responses. For the M.I. method, the default choice is the sampled response with the highest probability according to the marginalized pseudo joint distribution.

We also consider a version of the self-verification method of Kadavath et al. (2022) (denoted by S.V.) that, for a query $x$, first finds $Y_1$, the element with the largest (aggregated) probability (which is the default choice of S.E. method), and then calculates the probability of token *"True"* (normalized for the two tokens *"True"* and *"False"*) for the following query: *"Consider the following question: Q: $x$. One answer to question Q is $Y_1$. Is the above answer to question Q correct? Answer True or False. A:"*. The default choice of this baseline is the same as the default choice of the S.E. method. By this design, our intention is to construct a score that (unlike the first-order scores[9] we consider) is not sensitive to the size of the label set.

---

[8]In this context, the F1 score is calculated based on token inclusion (Joshi et al., 2017; Devlin et al., 2019): for two sequences $a = (a_1, \ldots, a_n)$ and $b = (b_1, \ldots, b_m)$, defining $p = |a \cap b|/n$ and $r = |a \cap b|/m$ (where $|a \cap b|$ is the size of the intersection of $a$ and $b$, in which for repetitions of an element $y$, we consider the minimum number of repetitions in $a$ and $b$, i.e., $\min_{c \in \{a,b\}} |\{i : c_i = y\}|$, in calculating the size of the intersection) we define $F1 = 2pr/(p + r)$. Relating to the standard definition of the F1 score, $p$ and $r$ play the role of precision and recall, respectively, if $a$ is thought of as a prediction of $b$.

[9]The scores $T0$ and S.E. are first order because they only consider the marginal distribution of a single response, unlike our uncertainty score which is based on MI estimation by considering (pseudo) joint distributions over multiple responses.

# H Experiments with Gemini 1.0 Nano-1 model

In order to show that our findings are not limited to models of certain size, we conducted experiments with much smaller Gemini Nano model with 2B parameters. The results are shown in Figures 7 and 8.

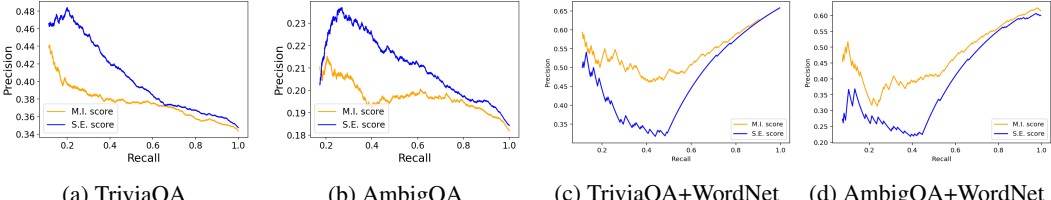

    (a) TriviaQA         (b) AmbigQA       (c) TriviaQA+WordNet    (d) AmbigQA+WordNet

Figure 7: The same figure as Figure 4, but for a much smaller Gemini Nano model with 2B parameters. PR-Curve for the baseline and the proposed methods on various datasets. On TriviaQA and AmbigQA datasets, S.E. outperforms M.I. when the recall is low, but they perform similarly as the recall increases (note that the large-looking difference on AmbigQA is actually at most 3%-points throughout, and it is also about at most the same for TriviaQA when the recall is above 0.4). For larger recalls, the two methods perform similarly, with the S.E. method somewhat outperforming our M.I. method. Note that the performance of the Nano model is quite weak, especially compared to the Gemini Pro results presented in the paper. On TriviaQA+WordNet and AmbigQA+WordNet datasets with the additional high entropy multi-label queries, M.I. outperforms S.E. baseline. Similarly to the experiments with Gemini Pro, the precision increases as the recall grows (above around 0.5), as the previously rejected WordNet data is accepted more and more.

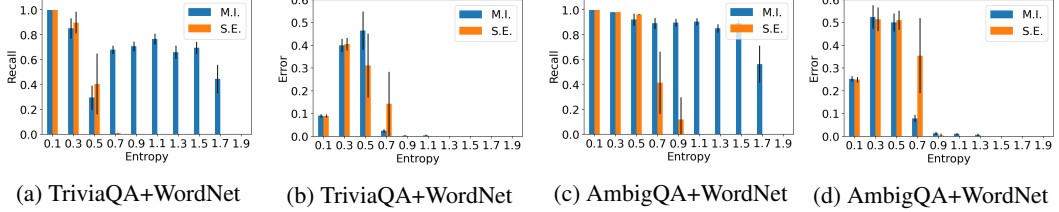

   (a) TriviaQA+WordNet   (b) TriviaQA+WordNet   (c) AmbigQA+WordNet   (d) AmbigQA+WordNet

Figure 8: Recall and error rates (percentage of mistakes when not abstaining) for TriviaQA and AmbigQA as a function of the entropy of model responses. The methods are calibrated at error rate 0.07 (as before, this error rate is computed by considering abstention as no error, i.e., error rate = errors/(predictions + abstentions)), based on 50 random samples. One can see that when the response entropy is small (the histogram is created with bins of withs 0.2), M.I. and S.E. have similar error and recall rates. On the other hand, for larger entropies, S.E. rejects all samples, while M.I. accepts some of them with a reasonably small error rate. This is again very similar to our findings for the Gemini Pro model.

