# OpenReview forum: "To Believe or Not to Believe Your LLM: Iterative Prompting for Estimating Epistemic Uncertainty"
_NeurIPS.cc/2024/Conference — NeurIPS 2024 poster_

### Official Review · Reviewer_1raE · 2024-07-12

**Soundness:** 3
**Presentation:** 2
**Contribution:** 3
**Rating:** 5
**Confidence:** 4

**Summary:**

This paper is about uncertainty quantification in LLMs, specifically illustrating some results based on an assumption involving independence of the distribution for a correct output for a question given some “iterative prompting”. The proposed information-theoretic measure is intended to help measure epistemic uncertainty of LLM output.

**Strengths:**

I think the paper has several interesting ideas, and one of the strengths is the attempt to formalize epistemic uncertainty using information theoretic ideas. There is novelty in the approach, in my view, as well as potential for these ideas to be useful in practice.

**Weaknesses:**

I found it generally hard to understand various sections of the paper, even though I am generally familiar with literature in this space. I think the writing is somewhat unclear in a few places. For instance, what is the scope of the work? Exactly what kinds of tasks are the proposed methods suitable for? Why should the key assumption hold and why should one trust it from some anecdotal examples?

Several statements do not seem fully explained or justified; I will mention a few later in my detailed comments. I’m also not sure that the paper does enough empirically to highlight the key ideas – if I’m mistaken, the authors should correct me. So, while I find the work interesting and potentially impactful, I feel there is room for improvement, particularly around the exposition. I am willing to revise my suggestion for the paper based on the discussion.

**Questions:**

Some comments and questions follow:

The title of the paper is too broad and could pretty much refer to any paper about uncertainty quantification in LLMs; I strongly recommend something more descriptive. One possibility is to use a sub-title such as “Iterative Prompting for Estimating Epistemic Uncertainty”.

I don’t understand why recent references have been used to cite epistemic vs. aleatoric uncertainty; these are older ideas. Either cite something classic or remove them.

The paragraph starting on line 47 makes some factual errors. The prior work does not assume there is a single correct response. For instance, in the Kuhn et al. paper, all answers that meet a threshold on the Rouge-L score (as compared to some ground truth) are deemed to be correct. I suggest rewriting this paragraph.

The “contributions” part on page 2 is quite unclear, and some things only become clearer later. The authors should write this in plain language that is understandable at a high level, without getting into details. For instance, what is the “ground truth language”?

Hallucinations are mentioned in the paper but never clearly defined or cited. I believe the notion here is different from that in other work.

I recommend using title case for section headers.

Fig. 1 is really hard to read and has inconsistent y-scales across panels. Also, I see a drop in the probability in some panels. How is this consistent with what is mentioned in the text? Is it because it does not go to 0 quicker? Please clarify.

I may be mistaken but I feel there may be something wrong with how Assumption 4.1 is written. It seems like Y_t must be a ground truth answer to the original question x (without the additional information in the prompt) but Y_1, through Y_t-1 can be any text. Is this true? The

How do we know that Assumption 4.1 is true? Is this based on the discussion in the preceding section? Does this depend on how long the ground truth response Y_t is?

Is F_0(x) defined? Perhaps this is the case where the prompt is: “Provide an answer to the following question: Q: x. A:”

I had a hard time figuring out what to make of the experiments. What is the main takeaway? When does the proposed approach perform better than baselines? There are other potential approaches (verbal and non-verbal, such as consistency-based approaches) that could be used as baselines. The P-R curves shown should mention coverage and accuracy; these are also known as accuracy-rejection curves.

**Limitations:**

Limitations do not seem to be discussed very much.

---

> ### Author Rebuttal · Authors · 2024-08-06
>
> We would like to thank reviewer for many insightful suggestions, we will revise the
> title, references, improve section 2.
>
> ## Scope, assumptions, and definitions
>
> * **Q:** What is the scope of the work? Exactly what kinds of
> tasks are the proposed methods suitable for?
> * **A:** Our main motivation was a question-answering systems where we
>   want to be able to detect whether answers to multiple-response
>   questions might be incorrect. Then, such systems can
>   abstain based on a high uncertainty score (Def. 4.4) after some calibration (see lines 219-229). Note that our proposed method is not suitable for
>   certification when the answer (or action) is correct with high
>   confidence.
>
> * **Q:** How do we know that Assumption 4.1 is true? Is this based on
>   the discussion in the preceding section? Does this depend on how
>   long the ground truth response $Y_t$ is?
> * **A:** The assumption is based on a common sense which we exemplify
> in this way: consider a query with single or multiple answers that
> could be encountered in the language (literature, internet, and other
> text sources). We assume that these answers are independent
> (i.e. occurrence of the answer in one textbook does not depend on
> the occurrence in another source). In fact, we will only care about
> such queries. This should be independent from how long $Y_t$ is.
>
> * **Q:** I feel there may be something wrong with how Assumption 4.1
> is written. It seems like $Y_t$ must be a ground truth answer to the
> original question $x$ (without the additional information in the prompt)
> but $Y_1$, through $Y_{t-1}$ can be any text. Is this true?
> * **A:** Thanks for pointing this out. Indeed, $Y_1, \ldots, Y_{t-1}$
>   are understood as a collection of random variables distributed
>   according to the ground truth (i.e. thet not arbitrary texts).
>
> * **Q:** Is $F_0(x)$ defined? Perhaps this is the case where the prompt
>   is: “Provide an answer to the following question: Q: x. A:”
> * **A:** Yes, you're correct. Thanks, we will define this.
>
> * **Comment:** Hallucinations are mentioned in the paper but never
>   clearly defined or cited. I believe the notion here is different
>   from that in other work.
> * **A:** Thanks for pointing out this oversight. In the context of our
> 	work a hallucination is a positive real quantity and it is
> 	captured by the KL divergence between pseudo-joint distributions
> 	of LLM and the ground-truth distribution respectively (Definition
> 	4.4).
>
> * **Q:** What is a ground-truth language?
> * **A:** In the context of our paper $X$ is a query, while $Y$ is a
> response, and ground truth $p$ is a stochastic model which captures
> relationships between queries and answers. In other words, it is a
> probabilistic model of language that we assume.
>
> ## Literature and Experiments
>
> * **Q:** Fig. 1 is really hard to read and has inconsistent y-scales
>   across panels. Also, I see a drop in the probability in some
>   panels. How is this consistent with what is mentioned in the text?
>   Is it because it does not go to 0 quicker? Please clarify.
> * **A:** The drop in probability is the behaviour we wanted to
>   pinpoint, that is, as the number of repetitions of incorrect answer
>   increases, the normalized probability drops. Indeed, it is
>   interesting that this varies for different prompts (sometimes drop
>   happens much faster). We suspect that this depends on the amount of
>   training data provided to the model, however it is hard to make
>   conclusions at this point and we leave this question as a future
>   research direction.
>
> * **Comment:** The paragraph starting on line 47 makes some factual
> errors. [...]  in the Kuhn et al. paper, all answers that meet a
> threshold on the Rouge-L score [...] are deemed to be correct.
> * **A:** Thanks for spotting this mistake, we will rewrite this
> accordingly. We meant this to be all semantically equivalent
> responses.
>
> * **Q:** I had a hard time figuring out what to make of the
>   experiments. What is the main takeaway? When does the proposed
>   approach perform better than baselines? There are other potential
>   approaches (verbal and non-verbal, such as consistency-based
>   approaches) that could be used as baselines. The P-R curves shown
>   should mention coverage and accuracy; these are also known as
>   accuracy-rejection curves.
> * **A:** Thanks for pointing out alternative baselines and
>   metrics. The goal of our experiments is to validate that a lower
>   bound on the uncertainty metric (theorem 4.5) indeed captured
>   incorrect answers in multiple-answer questions. We chose semantic
>   entropy as a reference since it is arguably a conceptually closest
>   baseline (since our lower bound is a mutual infromation of the
>   pseudo-joint distribution).

---

> > ### Comment · Reviewer_1raE · 2024-08-13
> > **Thanks for clarifications**
> >
> > I thank the authors for responding to many of my questions and for being willing to make some edits, such as title change, adding references, adding some definitions, etc. Although I find the paper to be empirically somewhat on the weaker side, I think there is some value to the literature from novel ideas. I will increase my score marginally in the expectation that exposition related issues will be suitably addressed in a revision.

---

> > > ### Author Response · Authors · 2024-08-13
> > >
> > > Thank you for your feedback, and for considering our response in your revised review.

---

### Official Review · Reviewer_DdzV · 2024-07-12

**Soundness:** 1
**Presentation:** 1
**Contribution:** 2
**Rating:** 3
**Confidence:** 4

**Summary:**

In the paper, the authors address the challenge of distinguishing between epistemic and aleatoric uncertainty in large language models (LLMs). They develop methods to decouple these uncertainties, which is crucial for handling queries with multiple valid responses. Their approach involves iterative prompting of the LLM to generate responses and measuring the sensitivity of new responses to previous ones. This helps in identifying cases where epistemic uncertainty is high, indicating potential hallucinations by the model. They propose an information-theoretic metric to quantify epistemic uncertainty and derive a computable lower bound for this metric, which is validated through experiments on datasets like TriviaQA and AmbigQA.

**Strengths:**

1. The topic is important and interesting.
2. The proposed iterative prompting strategy is practical and can be implemented easily.

**Weaknesses:**

1. The writing of this paper is poor and informal. The reviewer feels hard to understand the exact insights and the principles behind the paper. Please refer to Questions for a few of these confusions. The reviewer strongly recommends polishing the presentation carefully, especially those notations and definitions.
2. The evaluation protocol is unclear to the reviewer. In Fig. 6, what is the meaning of the entropy in the x-axis? Also, some LLM UQ baselines are missing such as [1]

Reference:

[1] Lin, Zhen, Shubhendu Trivedi, and Jimeng Sun. "Generating with confidence: Uncertainty quantification for black-box large language models." arXiv preprint arXiv:2305.19187 (2023).

**Questions:**

Section 2:
1.  "Moreover, consider a family of conditional distributions P ... ", if P is a set of distributions, why \mu is defined as a function? what is the meaning of \mu(x|x')?

2. what is "ground-truth conditional probability distribution"? do you mean p(Y|X) where (X, Y) are data and label?

3. where the "possible responses Y1, . . . , Yt" come from? Are they sampled from LLM Q with different decoding strategy?

4. what is the physical meaning of Z_i and also where does the second subscript i come from in (Z_i)_i?

Since the explanation of Z_i, \mu are missing, it is hard for the reviewer to identify how the "information-theoretical notations" contributes to this paper.

Section 3:

5. "To obtain conditional normalized probabilities, we consider the probabilities of the two responses, and normalize them so that they add to " it is unclear to the reviewer why this normalization is applied and how this is conducted. Please describe this procedure formally and justify the reason why conduct this.

6. The x-axis and y-axis labels are missing in Fig 1, Fig. 2, and Fig. 3.

7. How the proposed iterative prompting strategy (and its so-called "Conditional normalized probability") connected to epistemic uncertainty is missing. Although pieces of descriptions are mentioned in the Introduction, they are ambiguous and informal. Conventional definition of epistemic uncertainty is usually the model approximation error, e.g., p(\theta|D) where D is the training data. However, the new constructed input, i.e., the iterative prompt, is assumed to be significantly different from the D and how this is capable of quantify epistemic uncertainty is confusing for the reviewer.

**Limitations:**

No, the conclusion and limitations (and social impacts) are missing.

---

> ### Author Rebuttal · Authors · 2024-08-06
>
> ## Definition of epistemic uncertainty, iterative prompting, and connection between them
>
> * _**Q:** How the proposed iterative prompting strategy connected to
>   epistemic uncertainty is missing._
> * **A:** Note that the definition of epistemic uncertainty is given in
>   Definition 4.4. It is a KL-divergence between pseudo-joint
>   distribution (Def. 4.2) constructed by iteratively prompting LLM and a
>   pseudo-joint distribution of ground truth --- the formal connection
>   is introduced and discussed in detail in Section 4.  The main idea
>   of the paper is a lower bound on this quantity (Theorem 4.5) which
>   can be computed by only having access to LLM and iterative prompting
>   (constructing pseudo-joint distribution).
>
> * _**Q:** Conventional definition of epistemic uncertainty is usually
>   the model approximation error, e.g., $p(\theta|D)$ where $D$ is the
>   training data. However, the new constructed input, i.e., the
>   iterative prompt, is assumed to be significantly different from the
>   $D$ and how this is capable of quantify epistemic uncertainty is
>   confusing for the reviewer._
> * **A:** Note that $p(\theta | D)$ as you mention, is typically
> encountered in Bayesian literature, whereas we look at the problem
> from the frequentist viewpoint.  In our setting $\theta$ is not a
> random variable, but fixed throughout.  Note also that our metric of
> epistemic uncertainty and a theorem that suggests how to measure it
> do not make any assumptions on the training data. That said, we
> consider it as an advantage over methods where assumption on $D$ is
> required.
>
> * _**Q:** where the "possible responses $Y_1, \ldots , Y_t$" come from? Are
>   they sampled from LLM Q with different decoding strategy?_
> * **A:** Responses $Y_1, \ldots, Y_t$ are generated iteratively given
>   question $x$ exactly as described in the prompt in line 114. For
>   example, given question $x$, the first answer is $Y_1$; then given
>   $x, Y_1$, the second answer is $Y_2$; then we obtain $Y_3$ given $x,
>   Y_1, Y_2$, and so on. Kindly note that this is explained in Remark 4.3.
>
> ## Unclear definitions and notation
>
> * _**Q:** "Moreover, consider a family of conditional distributions
>   $\mathcal{P}$ ... ", if $\mathcal{P}$ is a set of distributions, why
>   $\mu$ is defined as a function? what is the meaning of $\mu(x|x')$?_
> * **A:** Here $\mu$ is indeed a discrete conditional probability
>   distribution, which is a function defined on $\mathcal{X}$ (the
>   space of finite text sequences) and that sums up to one. In other
>   words, $\mathcal{P}$ is a set of discrete distributions, one for $x'
>   \in \mathcal{X}$. We will make this clearer in the updated version.
>
> * _**Q:** what is "ground-truth conditional probability distribution"?
>   do you mean $p(Y|X)$ where $(X, Y)$ are data and label?_
> * **A:** In the context of our paper $X$ is a query, while $Y$ is a
> response, hence $p$ is a stochastic model which captures relationships
> between queries and answers. In other words, it is a probabilistic
> model of language that we assume.  In general, in supervised learning,
> $(X, Y)$ are typically understood as input and label.
>
> * **Q:** what is the physical meaning of $Z_i$ and also where does the
>   second subscript $i$ come from in $(Z_i)_i$ ?
> * **A:** $Z_1, Z_2, \ldots$ is a sequence of abstract random variables
>   with values in some measurable set $\mathcal{Z}$. These are simply
>   used to introduce some general definitions and avoid the use of $X$
>   and $Y$ which are reserved for questions and answers.  A shorthand
>   notation $(Z_i)_i$ is commonly used for describing tuples,
>   i.e. $(Z_i)_i = (Z_1, \ldots, Z_n)$.
>
> * _**Q:** Section 3: "To obtain conditional normalized probabilities,
>   we consider the probabilities of the two responses, and normalize
>   them so that they add to one." it is unclear to the reviewer why this
>   normalization is applied_
> * **A:** We apply normalization to have easy comparison between
> experiments with different number of repetitions of incorrect
> response.  In this way, it is easy to see that the factually correct
> answer decreases from $1$ to smaller probability as the number of
> repetitions increase.  Without normalization we would need to compare
> log-likelihoods, in which case the change would not be as obvious.
>
> ## Questions about evaluation protocol
>
> * _**Q:** In Fig. 6, what is the meaning of the entropy in the x-axis?_
> * **A:** In Fig. 6, the entropy on the x-axis represents the
> empirical entropy of the distribution of multiple answers obtained for
> a single query. We measure this for all queries and construct a
> histogram, by binning entropy values. This illustrates the frequency
> distribution of entropy across different queries, thereby providing
> insights into the variability and consistency of the responses.
> Specifically, empirical entropy measures the uncertainty or randomness
> in the distribution of these answers. Higher entropy values indicate
> greater diversity and less predictability among the answers, while
> lower entropy values suggest more uniform and predictable responses.
>
> * _**Comment:** Some LLM UQ baselines are missing such as (Zhen et al. 2023)._
> * **A:** Thank you for pointing this out, we will consider this
>   baselines in the updated version.

---

> > ### Comment · Area_Chair_GRRc · 2024-08-13
> > **Request for interaction**
> >
> > Dear Reviewer,
> >
> > I would appreciate if you could comment on the author's rebuttal, in light of the upcoming deadline.
> >
> > Thank you,
> > Your AC

---

> > ### Comment · Reviewer_DdzV · 2024-08-13
> >
> > Thank you for your response. The reviewer has thoroughly re-examined both the response and the clarified manuscript but still believes that significant revisions are necessary for the paper to be accepted. Therefore, the reviewer's evaluation remains unchanged.

---

> > > ### Author Response · Authors · 2024-08-13
> > >
> > > We strongly disagree that a significant revision is needed for the paper. Definitions, notations, evaluation protocol, etc. are already explained in detail in the paper, and we simply repeated the existing definitions in our rebuttal. Therefore, it is not clear what should be changed and why.

---

### Official Review · Reviewer_zm61 · 2024-07-13

**Soundness:** 4
**Presentation:** 4
**Contribution:** 3
**Rating:** 7
**Confidence:** 3

**Summary:**

The paper considers both epistemic and aleatoric uncertainties and proposes a novel method to decouple them. This method employs iterative prompting based on its previous responses. Experiments demonstrate that the proposed approach effectively detects cases where only epistemic uncertainty is large for multi-label questions.

**Strengths:**

1. Important problems: Aleatoric uncertainty is crucial for handling multi-label queries in practical applications.
2. Comprehensive theoretical analysis.
3. Demonstrates good performance in effectively dealing with multi-label queries by decoupling epistemic and aleatoric uncertainty.

**Weaknesses:**

1. **Limited application scope**: While I appreciate the method and performance of decoupling epistemic and aleatoric uncertainty for multi-label queries, it falls short in estimating uncertainty for single-label queries (Fig.5ab).

2. **Overlooked over-confidence issues**: The paper overlooks the problem of over-confidence, where the model produces low entropy among multiple responses despite providing incorrect answers.

3. **Limited dataset selection**: The paper filters the WordNet dataset, retaining only queries with entropy higher than 0.7. It claims that “both the proposed method and the semantic entropy method rarely make mistakes on this dataset, and therefore we are not adding any mistakes to either method.” This indicates that the selected dataset is relatively simple. However, as discussed in weakness 2, low entropy responses can still contain errors in more challenging datasets.

4. **Limited model selection**: The paper does not validate the effectiveness of the method across different types of models.

**Questions:**

N/A

**Limitations:**

The discussion on the limitations of this paper is insufficient.

---

> ### Author Rebuttal · Authors · 2024-08-07
>
> **Limited application scope:** on single-label queries, we should not expect to perform better than a competitive first-order method (such as S.E.), which is specifically designed for such queries. Please note that Fig.5ab in fact shows that our method performs essentially as well as the S.E. method in estimating uncertainty in single-label queries.
>
> **Overlooked over-confidence issues:** this is an important open question, and a solution will most likely require additional novel ideas.
>
> **Limited dataset selection:** given that TriviaQA and AmbigQA already contain challenging queries with low-entropy responses, we wanted to add queries with high entropy responses to demonstrate the limitations of first-order methods.
>
> **Limited model selection:** We have made similar observations on smaller models and with different architectures. We are in the process of delivering more results and will provide them during the discussion.

---

### Official Review · Reviewer_DRXq · 2024-07-13

**Soundness:** 3
**Presentation:** 3
**Contribution:** 2
**Rating:** 6
**Confidence:** 4

**Summary:**

This paper proposes an iterative prompt-based approach to uncertainty estimation. They make a model generate multiple answers, and estimate the probability of each, estimating uncertainty for each. They argue this method easily adapts to aleatoric and epistemic uncertainty, and can be applied to multiple choice question answering. They provide a mathematical framework to support their approach, arguing that a simple metric, Mutual Information between responses, can be a lower bound for uncertainty.

**Strengths:**

The epistemic/aleatoric approach to understanding uncertainty is a clear theoretic framework that is helpful in the MQA stage. This is highly appliable as threshodling methodology is provided, and complete examples are provided. Interpretability notions are used to explain this behaviour, giving an insight on why this happens, and not only how.

**Weaknesses:**

1) Assumption 4.1 is, to my understanding, that the correct answer will have a probability independently of "any" context. Under this assumption uncertainty is observed when the model relies more on context. Later math is based on this assumption, which seems somewhat task specific.While this works in the controlled question answering setup, context dependence holds a lot more information(ex:linguistic dependencies, uncertainty regarding user intent...). This could make the resulting uncertainty metric reduce less useful in deployment to general usage. I think discussion of limitation to assumption 4.1 should be more throughly discussed.
(It should be noted that 4.1/theorem 4.5 very clearly discusses the limitations of applying MI to infinite language)

2) Prompt based methods have been shown to be very model reliant, size reliant and tuning reliant. From a reproductibility perspective, it is unclear if results on Gemini will reproduce on different architectures, or smaller models. Is this behaviour a skill that emerges at a model size? requires specific tuning? or is it inherent to language modelling?
In the same line of questioning, it is unclear to me how the prompt was chosen, and if this selection procedure needs to be repeated on a new model. If this probabilistic approach is universal to Language modeling? If so this would truly be an interesting step forward from previous works. Otherwise it is a limitation.

**Questions:**

See weaknesses

**Limitations:**

Limitations are adressed.

---

> ### Author Rebuttal · Authors · 2024-08-07
>
> **Weaknesses 1:** Although the assumption is stated in this form for simplicity, the assumption in our theory is only for a very specific type of prompt we consider which still seeks answer to the original question $x$, and hence the ground-truth response should not change, and we also only apply it to specific contexts $Y_i$, i.e. those that could potentially be generated by the language as a response to the query. Thus, we do not really require independence for all contexts. Given this, we believe the assumption is not strong, and is intuitively stating that in the face of distractor responses, the response of the ground-truth language model should not change. Having said this, the formulation is indeed prone to adversarial attacks (and hence not perfect), e.g., if $Y_1=$"Answer1. However, we are interested in question $x_2$ instead of $x$", the new prompt would be
>
> "Consider the following question: Q: $x$. One answer to question Q is Answer1. However, we are interested in question $x_2$ instead of $x$. Provide an answer to the following question: Q: $x$"
>
> which could be misleading. However, we could change the assumption to only correspond to sequences $Y_i$ which are generated by a reasonable language model, and we will change it to restrict the family of $Y$'s (depending on $x$).
>
>
> **Weaknesses 2:** We believe this probabilistic approach is universal to language modeling. We have made similar observations on smaller models and with different architectures. We are in the process of delivering more results and will provide them during the discussion.
>
> We made almost no prompt engineering. The prompt used in the experiments was the second prompt tried (and not much different than the first try, which also gave very similar results).

---

> > ### Comment · Reviewer_DRXq · 2024-08-09
> >
> > Thank you for your response. I look forward to seeing those results. I am also taking some more time to digest your explanation on context, and will come back to you, I need to think it out.

---

> > ### Comment · Reviewer_DRXq · 2024-08-13
> >
> > Results promised on smaller models, other architectures, or other models in general were not provided (I completely understand those results are hard to provide in such a short timeline). Nonetheless despite your intuition that your results are universal, there is as of now no evidence.
> >
> > After thinking about your argument on weakness 1 for some time, I believe perhaps part of the issue is perhaps clarity? I did not understand what you're explaining here in the paper. I will leave my score as is, but pointing out that weaknesses I pointed out are not addressed, and are strong limitations that should be mentioned should this paper be accepted.

---

> > > ### Author Response · Authors · 2024-08-13
> > >
> > > We conducted additional experiments with a smaller language model, Gemini Nano, and the results are posted above as a comment to all reviewers. A link to an anonymised external page containing figures is shared with the area chair.
> > >
> > > Regarding weakness 1: We are not sure what explanations are missing, but we would be happy to provide additional clarifications in the remaining time.

---

### Author Rebuttal · Authors · 2024-08-07

We thank all reviewers for their insightful comments. We are in the process of delivering more results with different LLM architectures and sizes, and we aim to provide the results during the discussion. We have already observed that small LLM models behave similarly.

Please find our responses to other specific points below.

---

### Author Response · Authors · 2024-08-13
**Additional experimental results**

We conducted some additional experiments with a smaller language model, more precisely a 2B Gemini Nano model, which demonstrate that our findings with the much larger Gemini Pro models hold true in this much smaller scale.

Due to the heavy computational requirements, we could not finish the experiments in time to upload figures (the replications of Figures 5 and 6 from the paper for this model) during the rebuttal, and given the submission policies, we cannot share a link to an external page here. We give a summary of the results in tables below, and have also shared an anonymized link to the figures with the area chair, and hopefully it will be shared with the reviewers.

------------------------------------------------------------------------------------

Precision and recall results (Figure 5):


TriviaQA:
\begin{array}{l|c|c|c|c|c|}
\text{Recall} &0.2 & 0.4 & 0.6 & 0.8 & 1.0 \\\\
\hline
\text{Precision (M.I. score)} & 0.4 & 0.38 & 0.38 & 0.37 & 0.35 \\\\
\text{Precision (S.E. score)} & 0.47 & 0.42 & 0.39 & 0.38 & 0.35
\end{array}

AmbigQA:
\begin{array}{l|c|c|c|c|c|}
\text{Recall} & 0.2& 0.4& 0.6& 0.8& 1.0 \\\\
\hline
\text{Precision (M.I. score)} &0.205& 0.195& 0.2&  0.195& 0.185 \\\\
\text{Precision (S.E. score)} & 0.21& 0.225& 0.215& 0.2& 0.185
\end{array}

TriviaQA+WordNet:
\begin{array}{l|c|c|c|c|c|}
\text{Recall}& 0.2& 0.4& 0.6& 0.8& 1.0 \\\\
\hline
\text{Precision (M.I. score)}& 0.52& 0.47& 0.5& 0.55& 0.65 \\\\
\text{Precision (S.E. score)}& 0.45& 0.35& 0.45& 0.55& 0.65
\end{array}

AmbigQA+WordNet:
\begin{array}{l|c|c|c|c|c|}
\text{Recall}& 0.2& 0.4& 0.6& 0.8& 1.0 \\\\
\hline
\text{Precision (M.I. score)}& 0.35& 0.4& 0.45& 0.55& 0.62 \\\\
\text{Precision (S.E. score)}& 0.27& 0.23& 0.4& 0.55& 0.6
\end{array}

This is a replication of the results presented in Figure 5 in the paper, but for a much smaller Gemini Nano model with 2B parameters. On TriviaQA and AmbigQA datasets, S.E. outperforms M.I. when the recall is low, but they perform similarly as the recall increases (note that the difference on AmbigQA is at most 3%-points throughout, and it is also about at most the same for TriviaQA when the recall is above 0.4). For larger recalls, the two methods perform similarly, with the S.E. method somewhat outperforming our M.I. method. Note that the performance of the Nano model is quite weak, especially compared to the Gemini Pro results presented in the paper. On TriviaQA+WordNet and AmbigQA+WordNet datasets with the additional high entropy multi-label queries, M.I. outperforms the S.E. baseline. Similarly to the experiments presented in the paper, the precision increases as the recall grows (above around 0.5), as the previously rejected WordNet data is accepted more and more.

------------------------------------------------------------------------------------

Recall and error rates as functions of the entropy of the model's responses (Figure 6)

TriviaQA+WordNet:
\begin{array}{l|c|c|c|c|}
\text{Entropy bin}& 0.1& 0.3& 0.5& 0.7 \\\\
\hline
\text{M.I. Recall}& 0.3& 0.09& 0.09& 0.05 \\\\
\text{S.E. Recall}& 0.37& 0.0& 0.0& 0.0 \\\\
\text{M.I. Error}& 0.48& 0.2& 0.0& 0.0 \\\\
\text{S.E. Error}& 0.48& 0.0& 0.0& 0.0
\end{array}

AmbigQA+WordNet:
\begin{array}{l|c|c|c|c|}
\text{Entropy bin}& 0.1& 0.3& 0.5& 0.7 \\\\
\hline
\text{M.I. Recall}& 0.2& 0.05& 0.09& 0.05 \\\\
\text{S.E. Recall}& 0.2& 0.0& 0.0& 0.0 \\\\
\text{M.I. Error}& 0.6& 0.05& 0.0& 0.0 \\\\
\text{S.E. Error}& 0.6& 0.0& 0.0& 0.0
\end{array}

The methods are calibrated at error rate 0.07 (as in the paper, this error rate is computed by considering abstention as no error, i.e., error rate = #errors/(#predictions + #abstentions)), based on 50 random samples. One can see that when the response entropy is small (the histogram is created with bins of withs 0.2), M.I. and S.E. have similar error and recall rates. On the other hand, for larger entropies, S.E. rejects all samples, while M.I. accepts some of them with a reasonably small error rate. This is again very similar to our findings for the Gemini Pro model.

---

### Decision · Program_Chairs · 2024-09-25

**Decision:**

Accept (poster)

**Comment:**

The reviewers are split about this submission (3, 5, 6, 7).  While some reviewers have noted its significance, theoretical backing, intuitive algorithmic setup, and solid empirical results on the evaluated settings, one reviewer is strongly opposed to acceptance due to issues with clarity and formalization.  While I tend to agree that the writing is perhaps more technically advanced than that of a "regular" LLM paper, I have to disagree that it is inaccessible.

For this reason, I am leaning towards acceptance.